# Width-based Lookaheads with Learnt Base Policies and Heuristics Over the Atari-2600 Benchmark

**Stefan O'Toole**
Computing and Information Systems
University of Melbourne, Australia
stefan@student.unimelb.edu.au

**Nir Lipovetzky**
Computing and Information Systems
University of Melbourne, Australia
nir.lipovetzky@unimelb.edu.au

**Miquel Ramirez**
Electrical and Electronic Engineering
University of Melbourne, Australia
miquel.ramirez@unimelb.edu.au

**Adrian R. Pearce**
Computing and Information Systems
University of Melbourne, Australia
adrianrp@unimelb.edu.au

## Abstract

We propose new width-based planning and learning algorithms inspired from a careful analysis of the design decisions made by previous width-based planners. The algorithms are applied over the Atari-2600 games and our best performing algorithm, Novelty guided Critical Path Learning (N-CPL), outperforms the previously introduced width-based planning and learning algorithms $\pi$-IW(1), $\pi$-IW(1)+ and $\pi$-HIW(n, 1). Furthermore, we present a taxonomy of the Atari-2600 games according to some of their defining characteristics. This analysis of the games provides further insight into the behaviour and performance of the algorithms introduced. Namely, for games with large branching factors, and games with sparse meaningful rewards, N-CPL outperforms $\pi$-IW, $\pi$-IW(1)+ and $\pi$-HIW(n, 1).

## 1 Introduction

The Atari-2600 games provide useful environments for benchmarking autonomous agents due to the diversity of behaviour required across the different games. The Atari-2600 games can be accessed through the Arcade Learning Environment (ALE) [1] which provides a typical Reinforcement Learning (RL) environment interface where given a state, the agent selects an action and receives a resulting state and reward. The two main approaches that have been used by autonomous agents applied to the Atari-2600 games have been RL methods [2, 3, 4] and Planning methods [5, 6]. The RL approaches have had great success surpassing the performance of human players for many of the Atari-2600 games. However, RL approaches require long training times in order to train the Neural Networks (NN) used for policy and value functions. Planning agents do not require training time and instead use a bounded, fixed computational budget to decide which action to take at each time step of the game. The budget allowed for planning for each action is set as part of the experimental setting and can be set in such a way that the agent can play a game in real-time. Through the ALE interface, the agent is not provided a description of the transition or reward functions as is the case of models described through languages such as the Planning Domain Description Language (PDDL) [7]. Instead, planning agents applied to the Atari-2600 games are required to work with a simulator, treating the environment's transition and reward functions as a black-box [5].

Width-based planning agents have been shown to be particularly successful on the Atari-2600 games when compared to other planning agents [5, 6]. Width-based planners prioritise search effort on states deemed to be novel. The novelty of a state can be defined in a number of ways. Previously, novelty tests have been obtained from the RAM of the game [5], handcrafted features computed from screen

35th Conference on Neural Information Processing Systems (NeurIPS 2021).

pixels [6] and learnt features extracted from the screen pixels through a NN [8, 9, 10]. In this paper we consider planners with a novelty measure that does not require extensive feature engineering, or the internal state of the simulator, but is instead defined directly over the values of screen pixels.

Recent approaches have combined the RL and planning methods into single agents that are applied to the Atari-2600 games [8, 11, 10]. Junyent et al. [8] combined a width-based planner with a learnt policy defined over a NN in order to guide the planner to promising areas of the search space. The learnt NN was also used to extract features from which the novelty of states were defined over. In this paper we introduce new width-based planning and learning methods that learn both policy and value networks using a methodical learning schedule.

Through analysing previous width-based methods we construct and benchmark new width-based approaches for the Atari-2600 games. We also classify the Atari-2600 games according to their particular characteristics. The resulting game taxonomy helps us to gain insight into the performance of the algorithms we propose and benchmark. The paper contributions are: (1) an analysis of the previous width-based planning methods that have been applied to the Atari-2600 games, (2) introducing new width-based planning and learning approaches for playing the Atari-2600 games, (3) defining a methodical learning schedule for planning and learning methods, and (4) identifying characteristics of the Atari-2600 games that influence the performance of different planning approaches.

## 2 Background

### 2.1 MDPs

We model the Atari games as Markov Decision Processes (MDPs). We formalise MDPs, as described by Geffner and Bonet [12], as the tuple of $M = (S, s_0, A, T, R)$, where $S \subseteq \mathbb{R}^d$, $s_0 \in S$ is the initial state, $A$ provides the sets of applicable actions such that $A(s)$ is a set of actions applicable in $s \in S$, $T$ is a set of distributions such that $T(s, a, s')$ gives the probability of the transition from state $s \in S$ to state $s' \in S$ given action $a \in A(s)$, and $R$ is the reward function such that $R(s, a)$ returns the reward for performing action $a \in A(s)$ from state $s \in S$. In this work we will be considering a special case, finite-horizon MDPs, where accumulated rewards need to be maximised over a given number of stages $k = 1, \ldots, H$, starting at a fixed initial state, $s_0$. Terminal states in finite-horizon MDPs are absorbing states. That is, if $s$ is a terminal state and we are at time step $k$, every action $a$ will map $(s, k)$ into $(s, k + 1)$ and will be reward-free i.e. $R(s, a) = 0$. Our goal is to produce a policy, $\pi$, that maps any given state into an action, such that it maximises the expected accumulated reward received for an episode of the MDP,

$$\mathrm{argmax}_\pi E\left\{ \sum_{k=0}^{H-1} R(s_k, \pi(s_k)) \right\} \tag{1}$$

where the expectation is over $s_{k+1} \sim T(s_k, \pi(s_k), s_{k+1})$.

We assume that we have access to a simulator of the environment that given any state-action pair $(s, a)$, where $s \in S$ and $a \in A(s)$, the simulator returns the reward $R(s, a)$, a resulting state $s'$ following the probability distribution $T(s, a, s')$ and whether $s'$ is a terminal state. In line with previous work [6, 8, 10], we consider only *discrete* action sets and states that represent the internal state of the Atari environment such that action transitions are deterministic, that is $T(s, a, s')$ can only equal 1 for one state $s'$ and 0 for any other state $s''$, $s, s', s'' \in S$, $s' \neq s''$ and $a \in A(s)$. Note that while the agent can use the internal Atari game state to set the state of the simulator, it can only directly observe the Atari screen's pixel value for any given state.

### 2.2 Online Planning over simulators

In this paper we explore online planning over simulators by considering width-based lookahead algorithms for the Atari-2600 games. Lookaheads use a simulator of the environment to consider rewards from different action trajectories from the current state into the future. An example of this is shown in Figure 1 where a lookahead is illustrated. We define the notion of lookahead as,

**Definition 1** (Lookahead). A lookahead is defined as L = $(N, C, s_r)$ where $N$ is a set of nodes defined as state-action paths starting at the root state of the lookahead $s_r$, and $C$ is a function that given a node $n \in N$ and an action $a \in A(s)$ returns the children of $n$, that is $n_c \in C(n, a)$ and $n_c \in N$.

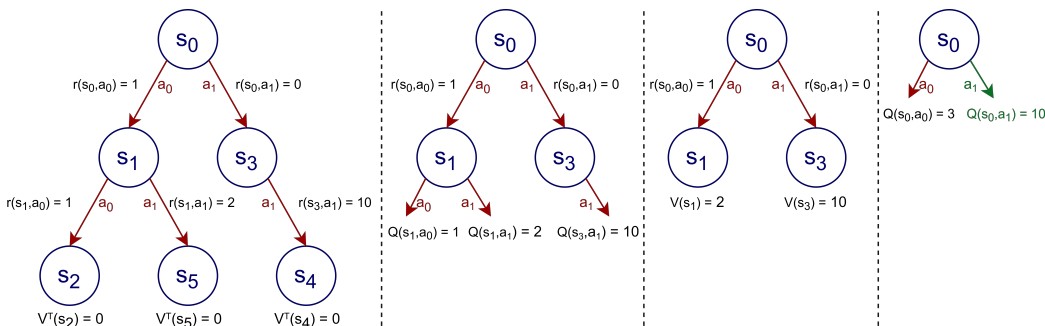

Figure 1: Example lookahead showing the action selection process for the root node $s_0$ following Definition 2. On the left, we have a fully built lookahead. From left to right, we show the recursive process to determine what is the action to be executed. The transition with action $a_1$ from $s_0$ shown in green in the last diagram on the right is the transition added to the lookahead's critical path (Definition 3). The $V$, $Q$, and $V^T$ functions are as defined in Definition 2.

Through backing up the rewards for each node in the lookahead, as shown in Figure 1, an expected value can be found for each action applicable at the current state. That is, where $n^s$ is the last state along the state-action path of node $n$, the operation of backing up the rewards and selecting which action to execute is,

**Definition 2** (Action selection of lookahead). Given a lookahead L= $(N, C, s_r)$ (Definition 1) the action to execute $a$ is selected at the root $n_r$, where $n_r^s = s_r$, by $\mathrm{argmax}_{a \in A(n_r^s)}\{Q(n_r, a)\}$, where $Q(n_r, a) = R(n_r^s, a) + \sum_{n \in C(n_r, a)} T(n_r^s, a, n^s)V(n)$, and $V(n) = V^T(n)$, with $V^T(n)$ being a termination cost, when $\bigcup_{a \in A(n^s)} C(n, a) = \emptyset$, otherwise $V(n) = \max_a\{R(n^s, a) + \sum_{n' \in C(n,a)} T(n^s, a, n'^s)V(n')\}$.

Once an action is selected for execution, the lookahead is updated to have its root at the selected action's resulting node and the lookahead continues being constructed from the new root node. We define the action selected by the agent as a part of its *critical path*. That is,

**Definition 3** (Critical Path). Given the action selected $a_t$ at each time step $t = 0, 1, \ldots, m$ following Definition 2, the critical path $\rho$ is the sequence of states and actions $\rho = (s_0, a_0, s_1, a_1, \ldots, a_{m-1}, s_m)$, such that $s_{i+1} \sim T(s_i, a_i, \cdot)$ for $i = 0, \ldots, m - 1$.

## 3 Related Work

### 3.1 MuZero

Schrittwieser et al. [11] followed up on the AlphaGo [13] and AlphaZero [14] algorithms with MuZero. AlphaZero is a planning and learning agent that achieved state-of-the-art performance on the games of Go, Chess and Shogi [15]. AlphaZero learns a policy and value network that are used within a Monte Carlo Tree Search (MCTS) lookahead through sampling actions according to the policy network and evaluating the states within the lookahead with the value function. Unlike AlphaZero, MuZero does not require a simulator or model of the game environment but instead learns a model of the environment through interaction. MuZero achieved state-of-the-art performance in the Atari-2600 games when compared to existing model-free RL algorithm performances. We also explore using learnt value and policy networks within a lookahead but consider width-based methods as opposed to MCTS. MuZero's experimental setting is different to the one considered in this paper as we require access to a simulator in the planning phase and use significantly less computing power.

### 3.2 Width-based Planners on Atari

Here we will provide an overview of the different width-based planners which have been applied to the Atari-2600 problems. In the next Section we go into the design and implementation details that each of the following planners use.

*Width-based planners* [16] prioritise adding states to the lookahead with novel valuations of features that are defined over the states. *IW(1)* is a width-based *breadth-first search* that is guaranteed to run in

*linear time and space* as it only expands *novel* states. *IW(1)* considers a state in the lookahead as *novel* if it is the *first* state within the lookahead to make a particular feature within the feature set true. Width-based planners were first applied to the Atari-2600 games by Lipovetzky et al. [5], where they applied IW(1) over the RAM values of the game state as features. Lipovetzky et al., and subsequent works that use IW(1) [17, 18], show that it outperforms breadth-first search and UCT [19] planners.

---

**Algorithm 1:** Overview of the RIW(1) Algorithm

**Input** : A lookahead $L = (N, C, s_r)$, and a base policy $\pi_b$
**Output** : Updated lookahead $L$

1 **while** $\neg$ `has_solved_label`$(s_r)$ **do**
2     $s \leftarrow s_r$ // complete depth-first rollout from the root node's state
3     **while** `is_novel`$(s) \wedge \neg$ `is_terminal`$(s)$ **do**
4        $s', a \leftarrow$ `sample_unsolved_child`$(s, \pi_b)$
5        $L \leftarrow$ `update_lookahead`$(L, s, a, s'), s \leftarrow s'$
6     **end**
7     `update_solved_labels`$(s)$
8 **end**

---

Bandres et al. [6] introduced a *depth-first* version of the IW(1) planner, named Rollout-IW(1) (RIW). RIW(1) aims to contain the same nodes that are expanded by the IW(1) planner. As RIW(1) performs depth-first search it was argued by Bandres et al. that it has better any-time performance than IW(1). The hypothesis for the better any-time performance of RIW(1) is that its search visits states that are further away from the initial state earlier in the search than its breadth-first search counter-part IW(1). Algorithm 1 provides an overview of RIW(1) using the base policy $\pi_b$. RIW(1) was originally defined to use a random uniform base policy $\pi_b$, however any given policy can be used instead. The function `sample_unsolved_child`, samples an action $a \sim \pi_b(s)$ and a transition $s' \sim T(s, a, \cdot)$ provided that $s'$ has not been marked as solved. If the selected transition does not already exist in the lookahead `update_lookahead` adds it. The `update_solved_labels` function adds a solved label to the given state and back-propagates the solved label to its parents if the parent's children have all been marked as solved. Bandres et al. results showed that RIW(1) outperformed IW(1) greatly when almost real-time budgets for planning were applied.

Junyent et al. [8] follow up on Bandres et al.'s [6] use of a random policy for the base policy ($\pi_b$) of RIW with $\pi$-IW(1), an algorithm that replaces $\pi_b$ with a trained policy defined over a NN. The intended effect is to orient the lookahead to promising areas of the state space. The NN from the trained policy is also used to extract features from the screen pixels for the computation of state novelty. Recently Junyent et al. [10] introduced $\pi$-IW(1)+ and $\pi$-HIW(n, 1) as follow ups of $\pi$-IW(1). $\pi$-IW(1)+ modifies $\pi$-IW(1)'s random breaking of ties for the action selection (Definition 2) to select the action with the branch of the lookahead that contains the most nodes. $\pi$-IW(1)+ also adds a learnt value function, $\tilde{V}$, which is used in the action selection (Definition 2) by modifying $V(s)$ to be $\max\{\tilde{V}(s), V^*(s)\}$, where $V^*(s)$ is $V(s)$ as described in Definition 2. $\pi$-HIW(n, 1) is a hierarchical algorithm that has a high-level planner which uses a coarse down-sampling of the screen pixels as a feature set and a low-level planner which uses $\pi$-IW(1)+ with the feature set defined through the policy network as previously described. The high-level planner uses a modified stochastic exploration policy, that selects actions with probability inversely proportional to state visitation counts.

## 4 Width-based Planning and Learning for the Atari Games

In this Section we step through different design considerations when constructing a width-based planning and learning algorithm. We compare the design decisions made by previous works and propose new algorithms to test over the Atari-2600 games.

### 4.1 Novelty Definitions: Classic and Depth-based

The novelty definition of a width-based lookahead dictates which states to prune. In Algorithm 1 the novelty definition determines the output of the `is_novel` function. We refer to IW(1)'s novelty definition as the "Classic" definition and define it as,

**Definition 4** ("Classic" Novelty). Given a feature set $F = \{f_1, \ldots, f_i, \ldots, f_k\}$ s.t. $f_i : S \rightarrow \{\top, \bot\}$, and a lookahead L= $(N, C, s_r)$, a node $n$ is novel, if $n$ contains the first state generated s.t. $f(n^s) = \top$ for some $f \in F$, that is, $\forall n' \in N, f(n'^s) = \bot$ and $n' \neq n$.

The "Depth" novelty definition introduced for RIW(1) by Bandres et al. [6] is,

**Definition 5** ("Depth" Novelty). Given a lookahead L= $(N, C, s_r)$, a newly generated node, $n \notin N$, reached after doing $d(n)$ actions from $s_r$, is novel, if $f(n^s) = \top$ for some $f \in F$, and $\forall n' \in N$, such that $d(n') \leq d(n), f(n'^s) = \bot$.

Later we show that in a width-based planning and learning algorithm based upon the RIW(1) algorithm the original "Classic" novelty is competitive and can sometimes outperform the depth-based one over the Atari-2600 games. In what follows, we refer to Bandres et al.'s original configuration of RIW(1) as $\text{RIW}_D$ and refer to RIW(1) where one replaces the "Depth" novelty definition with the "Classic" one as $\text{RIW}_C$.

## 4.2   Features for Novelty from Graphical Game Outputs

Width-based methods require a feature set $F$ to be defined over the observable state-space $S$. There are two types of observations that can be used for the Atari-2600 games, the internal states of the Atari-2600 machine (the RAM), and the colours of screen pixels. Either of these enable features to be defined, as arbitrary Boolean functions over the observable variables. For the internal state observables we have b $\times$ x variables, where b is the size of the Atari memory word (8 bits) and x is the size of the physical RAM given by the number of distinct memory addresses (128 addresses). There are c $\times$ w $\times$ h screen observable variables, where c is 128, w is 160, and h is 210, corresponding to the colour depth, and the number of screen pixels along the horizontal and vertical directions.

When RIW(1) was introduced [6], Bandres et al. stated that features capturing "meaningful structure" would yield better results than using raw features. Hence, Bandres et al. mapped the observable screen variables into the feature set B-PROST, first proposed by Liang et al. [3]. The B-PROST feature set attempts to capture temporal and spatial relationships between the past and present screen pixel values. In order to compute the set of B-PROST features there are a number of steps required. First a set of basic features needs to be computed through dividing the screen into $16 \times 14$ tiles comprised of $10 \times 15$ pixels. For each tile, $(w, h)$, where $w \in \{1, \ldots, 16\}$ and $h \in \{1, \ldots, 14\}$, there are $K$ features where $K$ is equal to the colour depth of the Atari-2600 pixels (128). The basic feature in the B-PROST set is $f_{w,h,c}$, where $c \in \{1, \ldots, K\}$ is true if the tile $(w, h)$ contains at least one pixel with the colour value $c$. A second tier of features, the Basic Pairwise Relative Offsets in Space (B-PROS) set, is computed from the basic ones. A B-PROS feature $f_{c_1,c_2,i,j}$, is true if $f_{w,h,c_1} \wedge f_{w+i,h+j,c_2}$ for any $w, h$. Finally, a third tier of features, the Basic Pairwise Relative Offsets in Time (B-PROT) set, are computed. A B-PROT feature considers the current screen's tiles $(w, h)$ and the previous game screen's tile $(w', h')$ so that a feature $f^t_{c_1,c_2,i,j}$ is true if $f_{w,h,c_1} \wedge f_{w'+i,h'+j,c_2}$ for any $w, h$ where $w' = w$ and $h' = h$. The B-PROST set is the union of basic, B-PROS, and B-PROT feature sets.

The feature set can also be defined dynamically through a NN [8]. $\pi$-IW, $\pi$-IW(1)+ and the lower level planner of $\pi$-HIW(n, 1) use a feature set that is defined as the output values of the rectified linear units from the last hidden layer of the policy NN treating zero values as $\bot$ and positive as $\top$. The policy NN input are the last four screens, processed to map colours to a suitably defined grayscale, and down sampled to a size of $84 \times 84$. While the policy network is being trained the features extracted through it will also change. This is similar to Dittadi et al. [9], who use Variational Autoencoders (VAE) to learn a set of features from the Atari game screen using a training set of game screens created from a RIW(1) execution using B-PROST. RIW(1) using the VAE features was shown to outperform RIW(1) using the B-PROST features.

While width-based planning methods using both the BPROST and NN extracted feature sets have been shown to perform well over the Atari games, previous width-based methods have not tested simpler feature sets defined directly over the screen pixel values. With the motivation of presenting a simpler width-based algorithm, we define our feature set directly over the values of the current down sampled $84 \times 84$, 8-bit grayscaled, observable screen variables. Each feature is defined as $f_{i,j,c}$ and is true if the downsampled pixel $(i, j)$ has the grayscaled colour c, where $i, j \in \{1, \ldots, 84\}$ and $c \in \{1, \ldots, 256\}$. Despite using a simpler feature set than previous work, in the next Section we show that our algorithm outperforms the methods that use dynamically defined NN based features.

**Algorithm 2:** Novelty guided Critical Path Learning (N-CPL)

```
  // Perform K training iterations
1 for i = 0, . . . , K do
2   | 𝒯ⁱ ← ∅, Eⁱ ← ∅ // Iteration's critical path transitions and episode rewards
3   | while ¬ train_interval_exhausted() do
4   |   | s ← s₀, R ← 0, L ← initialise_lookahead(s₀)
5   |   | while ¬ is_terminal(s) do
6   |   |   | L ←RIW(L, πᵦ) // Algorithm 1
7   |   |   | a, r, s′, L ← select_next_transition(L, Vᵗ) // Selected using Def.2
8   |   |   | 𝒯ⁱ ← 𝒯ⁱ ∪ (s, a, r, s′), s ← s′, R ← R + r
9   |   | end
10  |   | Eⁱ ← Eⁱ ∪ R // Episode rewards from current iteration
11  | end
    | // Train and update network parameters according to learning schedule
12  | πᵦ, Vᵗ ← update_network_parameters(𝒯ⁱ, Eⁱ)
13 end
```

### 4.3 Learning Base Policies and Termination Costs

AlphaGo [13] and $\pi$-IW [8] showed the power of using a learnt base policy defined through a NN in order to guide a lookahead search. Similarly, AlphaGo and $\pi$-IW(1)+ [10] also use a learnt value function defined through a NN. $\pi$-IW(1)+ used its learnt value function to modify the definition of $V(n)$ (Definition 2) allowing the rewards received from the transitions in the lookahead to sometimes be ignored in preference of the value network's valuation.

We propose a new algorithm based on Algorithm 1, Novelty guided Critical Path Learning or N-CPL for short, that incorporates both a learnt policy and value function network. An outline of N-CPL is shown in Algorithm 2. Like $\pi$-IW does, N-CPL defines the base policy used by Algorithm 1 to be a policy network. Besides that, N-CPL uses a cost-to-go approximation which we implement with a NN as a value function, which is evaluated at the non-terminal leaf nodes of the lookahead. Using cost-go-approximations has been shown to significantly improve the performance of width-based lookaheads over stochastic shortest paths [20], but rather than using simulations to obtain the cost-to-go estimates, we rely on a learnt heuristic function. That is, instead of assigning termination costs, $V^T$, of 0 as done by previous width-based methods, if the state is not terminal the valuation of the learnt value function network is used. We note that unlike $\pi$-IW(1)+ we do not use the learnt value function to modify the $V(s)$ definition as defined in Definition 2.

### 4.4 Learning from Critical Paths

The policy network of N-CPL uses the state action pairs, $(s_j, a_j)$ for $j = 0, \ldots, H - 1$, of previous episodes performed by N-CPL with NN parameters, $<\theta_i^\pi, \theta_i^V>$, in order to train new NN parameters $\theta_{i+1}^\pi$. This is similar to how $\pi$-IW(1) trains its policy function, except for the fact that $\pi$-IW(1) uses the Q values within the lookahead tree. If multiple actions in $\pi$-IW's lookahead have the same Q value for a given state, instead of the training vector assigning a probability of one to the executed action, $\pi$-IW uniformly distributes the probability across the actions with the same Q values. We have taken the simpler approach of just using 1-hot encodings for the single selected action along the critical path (Definition 3) of the N-CPL algorithm. Curating the training dataset in this way also means N-CPL does not need access to the internal data structures of the planning agent itself but instead can externally observe any agent interacting with the environment in order to acquire the training data.

Influential deep RL algorithms such as DQN [2] which have been applied to the Atari-2600 games rely on evaluating an $\epsilon$-greedy policy defined over the parameters of its network in order to sample transitions and use Q-learning updates on the parameters of the network. We follow this strategy and perform Temporal Differential (TD) [21] learning to train the value network. The selection of the transitions that are used for the TD learning determines what the value function is estimating. That is, TD's task is to estimate the expected accumulated rewards from the given state following the policy which underlies the transitions that it is trained on. In N-CPL the transitions within the

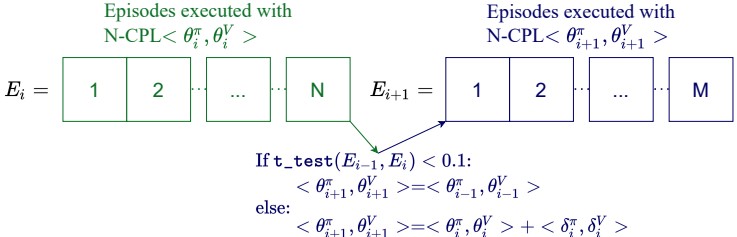

Figure 2: Illustration of the schedule for the network parameter updates, $<\theta^\pi, \theta^V>$, of N-CPL. The $E$ arrays contain episodes executed by N-CPL and each episode in the array is represented by a square in the diagram. The `t_test` function returns the p-value for Welch's t-test [22] of the episode rewards executed with the old $i-1$ parameters being better than the new ones. $\delta_i^\pi$ and $\delta_i^V$ are real vectors of the same dimensions as $\theta_i^\pi$ and $\theta_i^V$ respectively. $\delta_i^\pi$ and $\delta_i^V$ are functions over $E_i$.

lookahead follow the base policy, $\pi_b$ which is being learnt by aiming to mimic the policy induced from the N-CPL lookahead, $\pi_{\text{N-CPL}}$. The N-CPL lookahead through selecting actions according to Definition 2 can be seen as a policy improvement operator over $\pi_b$ and hence the execution of $\pi_{\text{N-CPL}}$ is not necessarily equivalent to the $\pi_b$. Therefore it does not make sense to approximate the expected accumulated reward of the lookahead with the expected accumulated reward of $\pi_b$. Instead, as shown in Line 8 of Algorithm 2, N-CPL trains only the transitions on the critical path of the lookahead (Definition 3), that is, the transitions selected by $\pi_{\text{N-CPL}}$. This results in the value function approximating the expected accumulated reward of executing $\pi_{\text{N-CPL}}$ from a given state.

## 4.5 Adding a Learning schedule

The previous width-based planning and learning methods continuously learn and update their policy and value networks, while a key mechanism of the Alpha-Zero algorithm [14] is the use of a learning schedule. Alpha-Zero evaluates each new set of network parameters $\theta'$ that are trained against the current set of network parameters $\theta$ to ensure $\theta'$ improves AlphaZero's performance. Here we introduce a general learning schedule mechanism that is applicable to sequentially executed and trained planning and learning methods applied to single-player domains, that N-CPL uses for updating its network parameters for its policy network, $\theta^\pi$ and value function network, $\theta^V$. As illustrated in Figure 2, the learning schedule determines whether network parameter updates $<\theta_i^\pi, \theta_i^V>$ can be accepted or if the $<\theta_{i-1}^\pi, \theta_{i-1}^V>$ parameters are kept, by evaluating their performance when used within N-CPL. This test is implemented in the `update_network_parameters` function shown in Algorithm 2. For the test N-CPL performs a Welch's t-test [22] on its performance with the new $i$ parameters vs. the old $i-1$ parameters. The update is rejected if the t-test suggests, with a $p$-value of less than 0.1, that performance could deteriorate if the new parameters were accepted. This training and parameter update schedule allows learning steps to be completed at each time step like done by $\pi$-IW [8], except that the updated parameters are not used for data generation by N-CPL until they have been accepted by the proposed learning schedule.

## 5 Experimental Study

We benchmark width-based planning and learning methods with variations of the design decisions explained in the previous section. Here we explain our experimental methodology and provide results across the different algorithms over the Atari-2600 games.

### 5.1 Methodology

Given vast compute resources it would be preferable to conduct a full ablation study over the Atari-2600 benchmark on each element of a width-based planning and learning algorithm discussed in the previous Section. However, due to computational constraints we instead select five planners which will provide the most insight. Two of the planners we evaluate are based on RIW(1) without learnt policy or value function networks. One of the planners uses the "Depth" definition of novelty (Definition 5) and the other uses the "Classic" definition (Definition 4), we name these version $\text{RIW}_D$

and $RIW_C$ respectively. Note that $RIW_D$ is as described in Bandres et al.[6], except that the features are defined directly over the screen's pixel values as discussed in the previous Section. The other 2 width-based planners we benchmark are two versions of N-CPL, as previously described. Again we test both the "Classic" and "Depth" novelty definitions (Definitions 4, 5), and refer to them as N-CPL and $N\text{-CPL}_D$ respectively. For the policy networks of N-CPL and $N\text{-CPL}_D$ we use the same architecture used by Mnih et al. [2]. The value network uses the same architecture as the policy network except that instead of the output layer being a dense softmax layer with an output for each action, the output layer of the value network is a dense linear layer with a single output value. Additionally we test a version of N-CPL that does not prune for novelty, we refer to as CPL, i.e. for CPL the `is_novel` function in Algorithm 1 always returns true.

We compare $RIW_D$, $RIW_C$, CPL, N-CPL, and $N\text{-CPL}_D$ to the $\pi$-IW, $\pi$-IW(1)+ and $\pi$-HIW(n, 1) planners with the results as given by Junyent et al. [10]. We do not directly compare our results to those given in the original RIW(1) planner as Bandres et al. [6] use a different experimental design. That is, Bandres et al. and previous works such as Lipovetzky et al. [5] benchmarked their width-based planners over the Atari-2600 games all using the full action set of 18 actions per state. We found in the code provided for the $\pi$-IW(1) work that it was benchmarked against the games using the minimal action set for each game. Using the minimal action set results in many games having much smaller branching factors, for example, instead of Breakout having a branching factor of 18, it has a branching factor of just 4. Additionally it is worth noting that the true average branching factor of each game is often much smaller than the minimal action set [23] and learning the minimal action set can help avoid unnecessary simulator interactions [18]. Due to computational constraints we could not benchmark our algorithms over the games with full, minimal and learnt minimal actions sets. Instead, we decided to benchmark using the minimal action set as Junyent et al. [10] do.

The results are also not directly compared with those from MuZero due to discrepancies in the evaluation protocols and computing resource requirements. For example, MuZero uses a smaller frameskip for the environment time steps and uses a longer allowed episode length of 108,000 frames compared to the 18,000 frame maximum episode length we impose. While our experiments run on a single vCPU for each trial for both training and evaluation, MuZero required 40 third generation Google Cloud TPUs for each run, 8 for training and 32 for its self-play. Furthermore the results for MuZero on each domain were only made available for a 20 billion frame training budget. We do however provide comparisons with the RL algorithms DQN [2] and Rainbow [4], along with all the necessary caveats about differences in evaluation protocols, in the Supplementary Material.

For the evaluation of each algorithm on each game we run 5 independent trials. Once training has completed we evaluate each trial over 10 episodes. Following previous width-based planning papers [6, 8, 10] we use a frameskip of 15. We keep our experimental settings the same as Junyent et al. [10] including a training budget of $2 \times 10^7$ simulator interactions and allowing 100 simulator interactions at each planning time step, which allows almost real-time planning. Note that previous width-based algorithms have varied in how they apply planning budgets, Lipovetzky et al. enforce a budget of 30,000 simulator interactions with a frameskip of 5, while Bandres et al. enforce time budgets of 0.5 and 32 seconds with a frameskip of 15. We ran 80 independent trials at once over 80 Intel Xeon 2.10GHz processors with 720GB of shared RAM, limiting each trial to run on a single vCPU. The average vCPU run-time per time step needed across both the planning and learning steps were 1.28 and 1.11 seconds for N-CPL, and $N\text{-CPL}_D$ respectively, resulting in each trial taking just under 3 days to complete. For $RIW_C$ and $RIW_D$, which do not require any learning steps or evaluation of NNs, the average run-times per step were 0.55 and 0.54 seconds respectively. Note that given Atari operates at 60 frames per second and we use a frameskip of 15 a real-time planner would be required to execute with a run-time of 0.25 seconds per time step.

The transitions within the lookahead are cached inline with previous width-based planners [5, 6, 8, 10]. That is, when the search revisits a transition between two nodes of the lookahead within the same episode, the simulator does not need to be recalled and hence does not affect the simulator budget. Also following previous work, transitions that are cached from previous time steps are not considered by the novelty Definitions 4, and 5, and hence will never be pruned.

## 5.2  Results

Table 1 summarises the results of N-CPL, $N\text{-CPL}_D$, CPL, $RIW_C$, $RIW_D$, $\pi$-IW(1), $\pi$-IW(1)+ and $\pi$-HIW(n, 1). Using the pairwise comparison of the different algorithms across the 53 games it is

Table 1: A pairwise comparison of the width-based planning algorithms over the full benchmark set made up of 53 Atari Games. Numbers represent the number of games an algorithm had a higher average evaluation score over the 5 learning trials than the algorithm it is being compared to.

| | Number of games with higher average score than | | | | | | | | |
| | N-CPL | N-CPL$_D$ | CPL | RIW$_C$ | RIW$_D$ | $\pi$-IW | $\pi$-IW+ | $\pi$-HIW | Total (ave. win %) |
|---|---|---|---|---|---|---|---|---|---|
| **N-CPL** | | 26 | 35 | 49 | 47 | 32 | 39 | 32 | **260 (70.1%)** |
| **N-CPL$_D$** | 26 | | 29 | 48 | 46 | 32 | 38 | 30 | **249 (67.1%)** |
| **CPL** | 18 | 24 | | 43 | 45 | 31 | 39 | 27 | **227 (61.2%)** |
| **RIW$_C$** | 3 | 4 | 10 | | 23 | 19 | 18 | 17 | **94 (25.3%)** |
| **RIW$_D$** | 5 | 6 | 8 | 29 | | 20 | 18 | 17 | **103 (27.8%)** |
| **$\pi$-IW** | 20 | 20 | 22 | 33 | 32 | | 30 | 25 | **182 (49.1%)** |
| **$\pi$-IW+** | 14 | 15 | 14 | 35 | 35 | 23 | | 23 | **159 (42.9%)** |
| **$\pi$-HIW** | 21 | 23 | 26 | 36 | 36 | 28 | 30 | | **200 (53.9%)** |

Table 2: Same as Table 1 but for games with a Branching Factor ≥ 10 (33 Games).

| | Number of games with higher average score than | | | | | | | | |
| | N-CPL | N-CPL$_D$ | CPL | RIW$_C$ | RIW$_D$ | $\pi$-IW | $\pi$-IW+ | $\pi$-HIW | Total (ave. win %) |
|---|---|---|---|---|---|---|---|---|---|
| **N-CPL** | | 14 | 22 | 30 | 29 | 23 | 28 | 22 | **168 (72.7%)** |
| **N-CPL$_D$** | 18 | | 18 | 31 | 29 | 23 | 27 | 21 | **167 (72.3%)** |
| **CPL** | 11 | 15 | | 26 | 28 | 23 | 28 | 21 | **152 (65.8%)** |
| **RIW$_C$** | 2 | 1 | 7 | | 16 | 15 | 16 | 13 | **70 (30.3%)** |
| **RIW$_D$** | 3 | 3 | 5 | 16 | | 16 | 16 | 13 | **72 (31.2%)** |
| **$\pi$-IW** | 9 | 9 | 10 | 17 | 16 | | 19 | 14 | **94 (40.7%)** |
| **$\pi$-IW+** | 5 | 6 | 5 | 17 | 17 | 14 | | 11 | **75 (32.5%)** |
| **$\pi$-HIW** | 11 | 12 | 12 | 20 | 20 | 19 | 22 | | **116 (50.2%)** |

clear that N-CPL$_D$ and N-CPL are the most performant. Comparing the "Depth" vs. "Classic" novelty definition methods as RIW$_D$ vs. RIW$_C$, the former performs better than the latter. The superiority of the "Depth" over the "Classic" definition of novelty does not follow when using our CPL method. The "Classic" method of N-CPL slightly outperforms the "Depth" method N-CPL$_D$, with Table 1 indeed showing that N-CPL is the best performing algorithm overall. Interestingly our CPL method that does not use novelty pruning, still outperforms all previous methods which shows the large contribution learning and using the policy and value function networks, as described in the previous Section, has on performance.

To better understand the performance of the algorithms we segment the benchmark set according to a couple of different characteristics. The game characteristics we examine are the *branching factor*, and the *sparseness* of meaningful reward feedback (SMRF). We consider rewards as meaningful when they provide information to a player about how to maximise the accumulated reward of an episode. For a given game, SMRF is determined by executing a random policy and a Real-Time Dynamic Planner (RTDP) [24] over each of the games. RTDP is an online planner that uses a one step lookahead (Definition 1) and an approximation for the termination cost $V^t$ at each of the leaf nodes. For the approximation of $V^t(s')$ we use the accumulated reward from a random policy executed from $s'$ for 10 time steps. We run both the random policy and RTDP for 50 time steps (750 frames). If the results from the RTDP planner are not better than the random policy according to Welch's t-test [22] with p < 0.1, the game is classified as having SMRFs. The results of the random policy vs. RTDP can be found in the Supplementary Material. For example, in the game of Pong, RTDP will be able to discover states through its 10 step approximation of $V^t(s')$ where either player has scored. Using information from $V^t(s')$ about which players have scored, RTDP will be able to have a better informed policy than the random policy, so Pong would not be considered a SMRF game. In a game like Skiing, where a skier is required to ski down a mountain and pass through gates on its path down, RTDP will not discover any meaningful rewards in its 10 step rollouts. This is because in Skiing, despite a constant negative reward at each time step, there is no meaningful reward feedback until the skier reaches the bottom of the mountain where a negative reward is applied for each gate the skier missed.

Table 2 groups the games according to their branching factor. Comparing Table 2 and 1 we can see that for games with larger branching factors, the relative performance gap between our N-CPL$_D$ and N-CPL planners and Junyent et al.'s $\pi$-IW(1), $\pi$-IW(1)+ and $\pi$-HIW(n, 1) increases as the branching factor increases. For example N-CPL and N-CPL$_D$ perform better than $\pi$-IW(1)+ in 28/33 (84.8%) games and 27/33 (81.8%) games respectively for the games with a branching factor greater or equal to 10. However, for games with a branching factor less than 10, N-CPL and N-CPL$_D$ only perform better than $\pi$-IW(1)+ in 11/20 (55%) and 11/21 (55%) games respectively.

Table 3: Same as Table 1 but for SMRF games (12 Games).

| | N-CPL | N-CPL$_D$ | CPL | RIW$_C$ | RIW$_D$ | $\pi$-IW | $\pi$-IW+ | $\pi$-HIW | Total (ave. win %) |
|---|---|---|---|---|---|---|---|---|---|
| | | | | Number of games with higher average score than | | | | | |
| **N-CPL** | | 7 | 5 | 10 | 10 | 7 | 9 | 7 | **55 (65.5%)** |
| **N-CPL$_D$** | 4 | | 3 | 9 | 9 | 7 | 9 | 6 | **47 (56%)** |
| **CPL** | 7 | 9 | | 11 | 11 | 8 | 9 | 7 | **62 (73.8%)** |
| **RIW$_C$** | 1 | 2 | 1 | | 6 | 6 | 3 | 4 | **23 (27.4%)** |
| **RIW$_D$** | 1 | 2 | 1 | 5 | | 6 | 3 | 4 | **22 (26.2%)** |
| **$\pi$-IW** | 4 | 4 | 4 | 5 | 5 | | 4 | 3 | **29 (34.5%)** |
| **$\pi$-IW+** | 3 | 3 | 3 | 9 | 9 | 8 | | 5 | **40 (47.6%)** |
| **$\pi$-HIW** | 5 | 6 | 5 | 8 | 8 | 9 | 7 | | **48 (57.1%)** |

Table 3 compares the pairwise performance for games classified as SMRF games. Table 3 clearly shows that the dominant method for the SMRF games is CPL, that is, the algorithm without novelty pruning. Table 3 also shows that the "Classic" novelty (Definition 4) methods outperform "Depth" novelty (Definition 5). These observations, that contradict previous claims in the literature, required careful analysis. We observed that the "Classic" method prunes states more aggressively, meaning it is more likely to reach states that are further away from the root node compared with the "Depth" definition. Similarly, as the CPL method does not prune any states due to novelty, CPL's depth-first lookahead trajectories will always reach the lookahead search horizon of 100 time steps at least once, given that the lookahead simulator budget is 100 time steps. This results in CPL on average searching for states that are further away from the root node than any of the novelty pruning methods. By definition, high rewards for SMRF games have a higher probability of being further away than games with dense rewards. Therefore, CPL and the "Classic" novelty methods are more likely to discover the meaningful rewards by searching deeper in the lookahead. Interestingly CPL and N-CPL still outperform, yet are close to $\pi$-IW(1)+ and $\pi$-HIW(n, 1) on the SMRF games, despite $\pi$-IW(1)+ and $\pi$-HIW(n, 1) being motivated by such domains.

# 6 Discussion

We have found significant discrepancies in the experimental settings used in the previous width-based planning papers for evaluating their algorithms over the Atari-2600 games. We believe a clear and consistent evaluation protocol should be set out for planning based algorithms applied to the Atari-2600 games to facilitate the direct comparison of their results. This could be similar to the evaluation protocol for the Atari-2600 games set out by Machado et al. [25], which was mainly focused towards RL agents and included recommendations on episode termination, setting of hyper-parameters, measuring training data, summarising learning performance and injecting stochasticity. However Machado et al. do not discuss evaluation settings that are vital to the deterministic planning setting we have explored in this paper, such as planning budgets, and caching of transitions within lookaheads. We hope that by having identified some of the discrepancies in the experimental settings of previous width-based algorithms, such as the size of the action set and the planning budget used, future research in planning agents for the Atari-2600 games can be more easily assessed. We were able to observe interesting patterns in the relative performance of algorithms through segmenting the Atari-2600 games by their different game characteristics. We are not aware of other works that analyse the performance of agents in regards to the characteristics of specific Atari-2600 games. We believe this taxonomy will provide useful insights into the behaviour of agents on the Atari-2600 games.

In this paper we have focused on width-based planning methods that have been applied over the Atari-2600 games. It is important to note though that these algorithms are defined in a general way to operate over MDPs. We proposed new width-based planning and learning algorithms through the examination of different design decisions made by previous implementations of width-based planners. These new algorithms, particularly N-CPL, are simpler implementations than the previously introduced width-based planning and learning algorithms $\pi$-IW(1)+ and $\pi$-HIW(n, 1). N-CPL defines its features directly over the grayscaled pixel colours of the game screen and uses a simplified novelty definition. Furthermore, N-CPL learns a value function which is only used for cost-to-go approximations at the leafs of the lookahead search tree. N-CPL also uses a methodical learning schedule we introduced for training both its policy and value networks. We found N-CPL to outperform $\pi$-IW(1), $\pi$-IW(1)+ and $\pi$-HIW(n, 1) not only across the Atari-2600 benchmark, but also over subsets of games with large branching factors and games with sparse meaningful rewards. These results show that N-CPL's integration of planning and learning pays off for almost real-time planning over hard problems.

## 7 Acknowledgements and Funding Disclosure

Funding in direct support of this work: Australian Government Research Training Program Scholarship provided by the Australian Commonwealth Government and the University of Melbourne; the Defence Science Institute, an initiative of the State Government of Victoria; and computing resources provided by the University of Melbourne through the Melbourne Research Cloud.

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
