# Additional results

Table 1: Comparing averages directly over 53 Atari-2600 games, as no confidence interval or standard deviation data is provided for the results of $\pi$-IW(1), $\pi$-IW(1)+, $\pi$-HIW(n, 1). The highest average score is highlighted in green. Freeway is excluded from this table as Junyent et al. [1] do not report the results for it due to its slow simulator time. See Table 2 for the Freeway results of $\text{RIW}_D$, $\text{RIW}_C$, $\text{N-CPL}_D$, and N-CPL.

| GAME | $\text{RIW}_D$ | $\text{RIW}_C$ | CPL | $\text{N-CPL}_D$ | N-CPL | $\pi$-IW | $\pi$-IW+ | $\pi$-HIW(n,1) |
|---|---|---|---|---|---|---|---|---|
| Alien | 4,365.20 | 4,478.40 | 6,209.00 | **7,640.60** | 6,943.40 | 3,969.78 | 2,585.77 | 4,609.18 |
| Amidar | 1,014.84 | 897.00 | 1,433.68 | **2,404.60** | 2,118.32 | 950.45 | 374.20 | 1,076.17 |
| Assault | 764.80 | 768.00 | **3,222.98** | 3,185.44 | 3,079.92 | 1,574.91 | 922.30 | 2,344.28 |
| Asterix | 52,940.00 | 54,090.00 | 39,860.00 | 48,364.00 | 48,226.00 | **346,409.11** | 247,063.36 | 90,017.25 |
| Asteroids | 1,480.20 | 1,397.20 | 6,145.20 | 9,000.80 | **9,152.80** | 1,368.55 | 1,490.87 | 990.95 |
| Atlantis | 48,930.00 | 46,464.00 | **173,786.00** | 120,650.00 | 119,636.00 | 106,212.63 | 143,177.73 | 17,539.22 |
| BankHeist | 453.46 | 436.90 | 336.40 | **957.12** | 709.00 | 567.16 | 256.29 | 501.68 |
| BattleZone | 102,780.00 | 88,560.00 | 220,680.00 | 165,340.00 | 153,880.00 | 69,659.40 | 30,848.95 | **309,137.79** |
| BeamRider | 4,124.37 | 3,521.80 | 6,164.64 | 3,743.80 | 3,560.88 | 3,313.11 | 8,428.96 | **11,931.41** |
| Berzerk | 600.00 | 620.00 | 3,148.00 | 4,642.20 | 4,120.60 | 1,548.23 | 960.03 | **7,417.26** |
| Bowling | 65.38 | 63.10 | **161.06** | 101.40 | 103.24 | 26.28 | 78.18 | 50.09 |
| Boxing | 52.44 | 54.58 | 80.96 | 84.30 | 86.40 | **99.88** | 88.19 | 6.81 |
| Breakout | 64.34 | 53.36 | 197.82 | **320.64** | 302.04 | 92.07 | 107.64 | 252.88 |
| Centipede | 52,685.12 | 55,495.78 | 53,608.80 | 60,157.38 | 62,654.16 | 126,488.35 | **141,070.19** | 80,685.48 |
| ChopperCommand | 3,768.00 | 3,466.00 | 17,908.00 | 4,570.00 | 3,786.00 | 11,187.44 | 3,431.74 | **70,787.12** |
| CrazyClimber | 40,520.00 | 39,387.50 | 78,266.00 | 90,332.00 | 91,912.00 | **161,192.01** | 138,648.58 | 102,205.99 |
| DemonAttack | 8,499.88 | 8,449.00 | 10,560.00 | 10,829.90 | 10,876.00 | 26,881.13 | **35,022.64** | 16,007.64 |
| DoubleDunk | 6.72 | 6.00 | 19.76 | 23.76 | **23.96** | 4.68 | -16.80 | 3.51 |
| Enduro | 1.90 | 1.34 | 231.44 | 250.88 | 220.28 | **506.59** | 63.83 | 44.47 |
| FishingDerby | -67.76 | -62.46 | -23.94 | -29.22 | -7.62 | **8.89** | -28.02 | -53.76 |
| Frostbite | 280.00 | 273.80 | **9,956.80** | 5,255.60 | 6,508.00 | 270.00 | 1,636.51 | 7,242.60 |
| Gopher | 6,311.43 | 5,990.83 | 11,181.60 | 13,326.40 | 12,539.60 | **18,025.91** | 7,061.76 | 15,001.18 |
| Gravitar | 1,755.00 | 1,725.00 | **2,382.00** | 2,246.00 | 2,284.00 | 1,876.80 | 1,532.33 | 1,154.01 |
| Hero | 17,438.30 | 17,159.70 | 29,200.40 | 34,083.40 | 36,099.80 | **36,443.73** | 22,097.39 | 36,231.21 |
| IceHockey | 21.48 | 21.92 | 19.62 | **29.58** | 26.96 | -9.66 | -4.02 | -2.36 |
| Jamesbond | 2,378.00 | 2,485.00 | 18,666.00 | 17,572.00 | **18,790.00** | 43.20 | 104.91 | 1,380.13 |
| Kangaroo | 1,556.00 | 1,504.00 | 5,332.00 | **9,222.00** | 9,202.00 | 1,847.46 | 2,918.98 | 6,861.57 |
| Krull | 2,176.58 | 2,127.60 | 5,631.22 | 4,900.90 | 4,422.88 | 8,343.30 | **13,014.77** | 4,121.81 |
| KungFuMaster | 5,668.00 | 5,886.00 | 27,288.00 | **43,520.00** | 42,388.00 | 41,609.03 | 24,871.94 | 20,680.65 |
| MontezumaRevenge | 0.00 | 0.00 | 2.00 | 0.00 | 0.00 | 0.00 | 810.49 | **5,275.89** |
| MsPacman | 15,697.56 | 15,729.28 | 11,922.22 | 16,150.08 | **18,285.18** | 14,726.33 | 5,916.86 | 4,523.47 |
| NameThisGame | 6,247.50 | 6,145.25 | 8,287.40 | 8,017.20 | 8,339.40 | 12,734.85 | **18,167.55** | 9,977.12 |
| Phoenix | 4,992.80 | 4,912.60 | 6,616.40 | 8,033.40 | **8,777.40** | 5,905.12 | 7,647.67 | 7,508.63 |
| Pitfall | -44.86 | -62.66 | -0.48 | -1.68 | **-0.42** | -214.75 | -2.46 | -128.82 |
| Pong | -4.52 | -4.92 | -12.28 | 6.18 | **10.94** | -20.42 | 2.14 | -9.70 |
| PrivateEye | 625.72 | 1,249.68 | 153.22 | 120.00 | 100.00 | 452.40 | 1,766.13 | **29,548.76** |
| Qbert | 4,499.50 | 4,426.50 | 28,182.50 | 31,625.50 | 30,618.50 | 32,529.60 | 23,337.90 | **40,449.72** |
| RoadRunner | 17,326.00 | 20,580.00 | 86,650.00 | 49,828.00 | 57,212.00 | 38,764.81 | 43,813.29 | **87,953.53** |
| Robotank | 31.03 | 31.33 | 31.04 | **37.94** | 36.16 | 15.66 | 9.68 | 10.63 |
| Seaquest | 1,609.60 | 1,576.20 | 3,523.00 | 2,878.60 | 3,922.80 | **5,916.05** | 559.28 | 867.51 |
| Skiing | -31,013.00 | -22,234.22 | -20,510.82 | -29,080.30 | -20,041.24 | -19,188.32 | **-13,852.04** | -15,417.86 |
| Solaris | 3,110.00 | 3,085.00 | **7,741.60** | 3,106.40 | 4,704.00 | 3,048.78 | 1,832.93 | 3,524.69 |
| SpaceInvaders | 2,592.40 | 2,622.50 | 3,447.40 | 3,680.40 | **4,289.40** | 2,694.09 | 1,622.49 | 2,946.18 |
| StarGunner | 17,510.00 | 17,597.56 | 18,340.00 | **20,700.00** | 20,320.00 | 1,381.24 | 1,642.82 | 1,864.64 |
| Tennis | **1.40** | -0.53 | -2.80 | 1.24 | 0.00 | -23.67 | -8.26 | -20.00 |
| TimePilot | 24,455.00 | 24,342.50 | 22,780.00 | 24,950.00 | 24,150.00 | 16,099.92 | 11,126.86 | **34,610.25** |
| Tutankham | 159.40 | 154.86 | 181.48 | 181.48 | 203.54 | **216.67** | 181.44 | 199.06 |
| UpNDown | 39,834.00 | 40,649.00 | 59,650.80 | 58,783.00 | 58,867.20 | **107,757.51** | 59,497.75 | 80,991.07 |
| Venture | 22.00 | 32.00 | **1,732.00** | 1,466.00 | 1,564.00 | 0.00 | 15.68 | 10.73 |
| VideoPinball | 136,531.50 | 138,748.35 | 148,995.14 | 134,489.58 | 139,799.22 | **514,012.51** | 387,308.60 | 184,720.01 |
| WizardOfWor | 26,956.25 | 27,991.11 | 54,988.00 | 43,556.00 | 43,436.00 | **76,533.18** | 30,383.68 | 12,027.43 |
| YarsRevenge | 59,779.48 | 60,940.90 | 133,647.42 | 142,568.26 | 135,089.68 | 102,183.67 | 64,544.51 | **159,496.20** |
| Zaxxon | 9,342.00 | 9,520.00 | 28,102.00 | **33,268.00** | 30,818.00 | 22,905.73 | 10,159.01 | 21,135.58 |
| **Total times best** | **1** | **0** | **7** | **10** | **8** | **12** | **5** | **10** |

Table 2: Average scores with 90% confidence intervals over the set of 54 Atari Games. Algorithm scores that are the best according to the Welch's t-test [2] using p< 0.1 are highlighted in green.

| GAME | $RIW_D$ | $RIW_C$ | CPL | $N\text{-}CPL_D$ | N-CPL |
|---|---|---|---|---|---|
| Alien | 4,365.20±471.95 | 4,478.40±375.51 | 6,209.00±415.77 | 7,640.60±615.52 | 6,943.40±507.28 |
| Amidar | 1,014.84±70.76 | 897.00±70.28 | 1,433.68±123.49 | **2,404.60±69.72** | 2,118.32±114.28 |
| Assault | 764.80±49.88 | 768.00±68.03 | 3,222.98±98.37 | 3,185.44±123.18 | 3,079.92±122.33 |
| Asterix | 52,940.00 ±1,486.56 | 54,090.00 ±1,323.40 | 39,860.00 ±410.25 | 48,364.00 ±1,844.10 | 48,226.00 ±1,184.51 |
| Asteroids | 1,480.20±75.48 | 1,397.20±113.97 | 6,145.20±318.23 | 9,000.80±143.15 | 9,152.80±189.24 |
| Atlantis | 48,930.00 ±4,017.54 | 46,464.00 ±3,132.23 | **173,786.00 ±2,343.28** | 120,650.00 ±892.42 | 119,636.00 ±830.95 |
| BankHeist | 453.46±31.86 | 436.90±40.51 | 336.40±25.25 | **957.12±105.73** | 709.00±76.09 |
| BattleZone | 102,780.00 ±22,127.98 | 88,560.00 ±16,866.49 | **220,680.00 ±14,274.79** | 165,340.00 ±28,612.76 | 153,880.00 ±28,305.50 |
| BeamRider | 4,124.37±384.99 | 3,521.80±371.92 | **6,164.64±287.17** | 3,743.80±466.05 | 3,560.88±415.25 |
| Berzerk | 600.00±32.84 | 620.00±32.37 | 3,148.00±386.38 | 4,642.20±367.59 | 4,120.60±423.64 |
| Bowling | 65.38±1.66 | 63.10±2.00 | **161.06±5.46** | 101.40±3.22 | 103.24±3.32 |
| Boxing | 52.44±2.00 | 54.58±2.86 | 80.96±2.63 | 84.30±2.10 | 86.40±0.97 |
| Breakout | 64.34±17.89 | 53.36±10.91 | 197.82±33.59 | 320.64±8.03 | 302.04±18.31 |
| Centipede | 52,685.12 ±2,811.22 | 55,495.78 ±1,398.86 | 53,608.80 ±1,567.38 | 60,157.38 ±1,726.88 | **62,654.16 ±1,107.16** |
| ChopperCommand | 3,768.00±634.47 | 3,466.00±378.06 | **17,908.00±1,432.95** | 4,570.00±602.78 | 3,786.00±601.16 |
| CrazyClimber | 40,520.00 ±550.63 | 39,387.50 ±486.25 | 78,266.00 ±3,572.79 | 90,332.00 ±2,414.91 | 91,912.00 ±2,184.99 |
| DemonAttack | 8,499.88±252.38 | 8,449.00±320.73 | 10,560.00±188.97 | 10,829.90±274.19 | 10,876.00±193.47 |
| DoubleDunk | 6.72±0.84 | 6.00±0.94 | 19.76±1.02 | 23.76±0.15 | **23.96±0.07** |
| Enduro | 1.90±0.57 | 1.34±0.38 | 231.44±10.04 | **250.88±8.23** | 220.28±9.71 |
| FishingDerby | -67.76±1.96 | -62.46±2.20 | -23.94±4.32 | -29.22±4.30 | **-7.62±5.99** |
| Freeway | 5.50 ± 0.24 | 5.52 ± 0.28 | 28.86±0.45 | 29.02 ± 0.45 | 28.96 ± 0.30 |
| Frostbite | 280.00±4.56 | 273.80±3.72 | **9,956.80±1,066.62** | 5,255.60±389.33 | 6,508.00±588.79 |
| Gopher | 6,311.43±440.94 | 5,990.83±479.70 | 11,181.60±72.70 | **13,326.40±240.79** | 12,539.60±158.58 |
| Gravitar | 1,755.00±171.35 | 1,725.00±192.75 | 2,382.00±228.46 | 2,246.00±201.67 | 2,284.00±231.37 |
| Hero | 17,438.30 ±1,011.92 | 17,159.70 ±949.20 | 29,200.40 ±1,804.42 | 34,083.40 ±938.86 | **36,099.80 ±553.32** |
| IceHockey | 21.48±0.73 | 21.92±0.83 | 19.62±1.09 | **29.58±0.90** | 26.96±1.03 |
| Jamesbond | 2,378.00±1,224.25 | 2,485.00±1,210.76 | 18,666.00±598.37 | 17,572.00±820.54 | 18,790.00±907.58 |
| Kangaroo | 1,556.00±203.70 | 1,504.00±151.18 | 5,332.00±727.16 | 9,222.00±546.79 | 9,202.00±572.19 |
| Krull | 2,176.58±89.61 | 2,127.60±88.85 | **5,631.22±272.50** | 4,900.90±141.45 | 4,422.88±149.38 |
| KungFuMaster | 5,668.00 ±425.75 | 5,886.00 ±521.52 | 27,288.00 ±1,908.35 | 43,520.00 ±1,108.67 | 42,388.00 ±643.13 |
| MontezumaRevenge | 0.00±0.00 | 0.00±0.00 | 2.00±3.26 | 0.00±0.00 | 0.00±0.00 |
| MsPacman | 15,697.56 ±1,148.98 | 15,729.28 ±1,138.48 | 11,922.22 ±1,044.99 | 16,150.08 ±1,064.72 | **18,285.18 ±703.36** |
| NameThisGame | 6,247.50±98.52 | 6,145.25±91.49 | 8,287.40±117.61 | 8,017.20±129.51 | 8,339.40±109.81 |
| Phoenix | 4,992.80±242.42 | 4,912.60±262.97 | 6,616.40±316.44 | 8,033.40±687.80 | 8,777.40±776.53 |
| Pitfall | -44.86±22.53 | -62.66±26.84 | -0.48±0.78 | -1.68±1.33 | -0.42±0.52 |
| Pong | -4.52±1.49 | -4.92±1.28 | -12.28±1.29 | 6.18±1.29 | **10.94±1.22** |
| PrivateEye | 625.72±685.72 | 1,249.68±938.31 | 153.22±22.74 | 120.00±9.30 | 100.00±0.00 |
| Qbert | 4,499.50±756.07 | 4,426.50±688.09 | 28,182.50±2,855.33 | **31,625.50±406.22** | 30,618.50±693.66 |
| RoadRunner | 17,326.00 ±3,626.95 | 20,580.00 ±3,760.02 | **86,650.00 ±3,065.43** | 49,828.00 ±3,244.08 | 57,212.00 ±4,094.90 |
| Robotank | 31.03±1.16 | 31.33±1.33 | 31.04±0.63 | **37.94±0.73** | 36.16±0.84 |
| Seaquest | 1,609.60±291.85 | 1,576.20±281.51 | 3,523.00±255.00 | 2,878.60±285.43 | **3,922.80±245.47** |
| Skiing | -31,013.00 ±700.82 | -22,234.22 ±872.31 | -20,510.82 ±1,199.55 | -29,080.30 ±1,034.25 | -20,041.24 ±1,139.52 |
| Solaris | 3,110.00±232.31 | 3,085.00±532.56 | **7,741.60±825.51** | 3,106.40±169.24 | 4,704.00±615.29 |
| SpaceInvaders | 2,592.40±301.45 | 2,622.50±303.10 | 3,447.40±269.92 | 3,680.40±310.85 | **4,289.40±301.15** |
| StarGunner | 17,510.00±240.75 | 17,597.56±397.74 | 18,340.00±217.12 | 20,700.00±178.37 | 20,320.00±335.94 |
| Tennis | 1.40±1.53 | -0.53±1.52 | -2.80±0.93 | 1.24±1.45 | 0.00±1.51 |
| TimePilot | 24,455.00±615.53 | 24,342.50±544.82 | 22,780.00±839.59 | 24,950.00±528.00 | 24,150.00±670.57 |
| Tutankham | 159.40±4.77 | 154.86±4.86 | 184.94±4.59 | 181.48±3.80 | **203.54±4.82** |
| UpNDown | 39,834.00±830.87 | 40,649.00±822.57 | 59,650.80±1,078.03 | 58,783.00±604.10 | 58,867.20±752.05 |
| Venture | 22.00±20.96 | 32.00±23.00 | **1,732.00±49.99** | 1,466.00±176.15 | 1,564.00±149.66 |
| VideoPinball | 136,531.50 ±8,459.01 | 138,748.35 ±8,349.10 | 148,995.14 ±8,492.72 | 134,489.58 ±7,483.99 | 139,799.22 ±7,019.84 |
| WizardOfWor | 26,956.25 ±2,793.49 | 27,991.11 ±2,907.59 | **54,988.00 ±1,906.42** | 43,556.00 ±4,275.90 | 43,436.00 ±4,142.91 |
| YarsRevenge | 59,779.48 ±2,070.84 | 60,940.90 ±2,234.00 | 133,647.42 ±4,698.49 | **142,568.26 ±3,934.50** | 135,089.68 ±4,813.68 |
| Zaxxon | 9,342.00 ±944.44 | 9,520.00 ±1,221.31 | 28,102.00 ±1,264.90 | **33,268.00 ±1,192.32** | 30,818.00 ±1,722.99 |
| **Best algorithm (t-test, p <0.1)** | 0 | 0 | 11 | 9 | 9 |

Table 3: Comparison of experimental settings used for the results of the different algorithms. Note that the train budget includes all the simulator interactions used by the lookahead algorithms even though only a small fraction of simulator interactions are used for training the networks directly.

| Algorithm | Frameskip | Max. ep. length (Frames) | Train Budget (Sim. Interactions) | Lookahead Budget (Sim. Interactions) | Starts | Loss of Life signal |
|---|---|---|---|---|---|---|
| $\mathbf{RIW}_D$, $\mathbf{RIW}_C$ | 15 | 18,000 | 0 | 100 | - | No |
| $\mathbf{N\text{-}CPL}_D$, $\mathbf{N\text{-}CPL}$, $\mathbf{CPL}$, $\pi\text{-}\mathbf{IW}$, $\pi\text{-}\mathbf{IW+}$, $\pi\text{-}\mathbf{HIW(n,1)}$ | 15 | 18,000 | $20 \times 10^6$ | 100 | - | No |
| **DQN** | 4 | 18,000 | $50 \times 10^6$ | NA | Rand.no-ops | Yes |
| **Rainbow** | 4 | 108,000 | $50 \times 10^6$ | NA | Human and Rand. no-ops | Yes |

Table 4: Comparison of N-CPL with a Human player's scores and the model-free RL algorithm DQN scores as reported by Mnih et al.[3]. Note that the experimental settings are different between N-CPL and DQN[3], in terms of training budget, frame skips, using no-op starts and loss of life signal, see Table 3 for a comparison of experimental settings.

| GAME | Human | DQN | N-CPL |
|---|---|---|---|
| Alien | 6,875.00 | 3,069.00 | **6,943.40** |
| Amidar | 1,676.00 | 739.50 | **2,118.32** |
| Assault | 1,496.00 | **3,359.00** | 3,079.92 |
| Asterix | 8,503.00 | 6,012.00 | **48,226.00** |
| Asteroids | **13,157.00** | 1,629.00 | 9,152.80 |
| Atlantis | 29,028.00 | 85,641.00 | **119,636.00** |
| BankHeist | **734.40** | 429.70 | 709.00 |
| BattleZone | 37,800.00 | 26,300.00 | **153,880.00** |
| BeamRider | 5,775.00 | **6,846.00** | 3,560.88 |
| Bowling | **154.80** | 42.40 | 103.24 |
| Boxing | 4.30 | 71.80 | **86.40** |
| Breakout | 31.80 | **401.20** | 302.04 |
| Centipede | 11,963.00 | 8,309.00 | **62,654.16** |
| ChopperCommand | **9,882.00** | 6,687.00 | 3,786.00 |
| CrazyClimber | 35,411.00 | **114,103.00** | 91,912.00 |
| DemonAttack | 3,401.00 | 9,711.00 | **10,876.00** |
| DoubleDunk | -15.50 | -18.10 | **23.96** |
| Enduro | **309.60** | 301.80 | 220.28 |
| FishingDerby | **5.50** | -0.80 | -7.62 |
| Freeway | 29.60 | **30.30** | 28.96 |
| Frostbite | 4,335.00 | 328.30 | **6,508.00** |
| Gopher | 2,321.00 | 8,520.00 | **12,539.60** |
| Gravitar | **2,672.00** | 306.70 | 2,284.00 |
| Hero | 25,763.00 | 19,950.00 | **36,099.80** |
| IceHockey | 0.90 | -1.60 | **26.96** |
| Jamesbond | 406.70 | 576.70 | **18,790.00** |
| Kangaroo | 3,035.00 | 6,740.00 | **9,202.00** |
| Krull | 2,395.00 | 3,805.00 | **4,422.88** |
| KungFuMaster | 22,736.00 | 23,270.00 | **42,388.00** |
| MontezumaRevenge | **4,367.00** | 0.00 | 0.00 |
| MsPacman | 15,693.00 | 2,311.00 | **18,285.18** |
| NameThisGame | 4,076.00 | 7,257.00 | **8,339.40** |
| Pong | 9.30 | **18.90** | 10.94 |
| PrivateEye | **69,571.00** | 1,788.00 | 100.00 |
| Qbert | 13,455.00 | 10,596.00 | **30,618.50** |
| Riverraid | 13,513.00 | 8,316.00 | **22,111.20** |
| RoadRunner | 7,845.00 | 18,257.00 | **57,212.00** |
| Robotank | 11.90 | **51.60** | 36.16 |
| Seaquest | **20,182.00** | 5,286.00 | 3,922.80 |
| SpaceInvaders | 1,652.00 | 1,976.00 | **4,289.40** |
| StarGunner | 10,250.00 | **57,997.00** | 20,320.00 |
| Tennis | -8.90 | -2.50 | **0.00** |
| TimePilot | 5,925.00 | 5,947.00 | **24,150.00** |
| Tutankham | 167.60 | 186.70 | **203.54** |
| UpNDown | 9,082.00 | 8,456.00 | **58,867.20** |
| Venture | 1,188.00 | 380.00 | **1,564.00** |
| VideoPinball | 17,298.00 | 42,684.00 | **139,799.22** |
| WizardOfWor | 4,757.00 | 3,393.00 | **43,436.00** |
| Zaxxon | 9,173.00 | 4,977.00 | **30,818.00** |
| # Games >human | | 23 (47%) | **37 (76%)** |
| #Games >75% human | | 27 (55%) | **40 (82%)** |
| # Games Best | 10 (20%) | 8 (16%) | **31 (63%)** |

Table 5: Comparison of N-CPL with a Human player's scores and the model-free RL algorithm Rainbow using human starts. Note that the experimental settings are different between N-CPL and Rainbow[4], in terms of the training budget, maximum episode length, frame skips, using human starts and loss of life signal, see Table 3 for a comparison of experimental settings.

| GAME | Human | Rainbow | N-CPL |
|---|---|---|---|
| Alien | 6875 | 6022.90 | **6943.40** |
| Amidar | 1676 | 202.80 | **2118.32** |
| Assault | 1496 | **14491.70** | 3079.92 |
| Asterix | 8503 | **280114.00** | 48226.00 |
| Asteroids | **13157** | 2249.40 | 9152.80 |
| Atlantis | 29028 | **814684.00** | 119636.00 |
| BankHeist | 734.4 | **826.00** | 709.00 |
| BattleZone | 37800 | 52040.00 | **153880.00** |
| BeamRider | 5775 | **21768.50** | 3560.88 |
| Bowling | **154.8** | 39.40 | 103.24 |
| Boxing | 4.3 | 54.90 | **86.40** |
| Breakout | 31.8 | **379.50** | 302.04 |
| Centipede | 11963 | 7160.90 | **62654.16** |
| ChopperCommand | 9882 | **10916.00** | 3786.00 |
| CrazyClimber | 35411 | **143962.00** | 91912.00 |
| DemonAttack | 3401 | **109670.70** | 10876.00 |
| DoubleDunk | -15.5 | -0.60 | **23.96** |
| Enduro | 309.6 | **2061.10** | 220.28 |
| FishingDerby | 5.5 | **22.60** | -7.62 |
| Freeway | **29.6** | 29.10 | 28.96 |
| Frostbite | 4335 | 4141.10 | **6508.00** |
| Gopher | 2321 | **72595.70** | 12539.60 |
| Gravitar | **2672** | 567.50 | 2284.00 |
| Hero | 25763 | **50496.80** | 36099.80 |
| IceHockey | 0.9 | -0.70 | **26.96** |
| Kangaroo | 3035 | **10841.00** | 9202.00 |
| Krull | 2395 | **6715.50** | 4422.88 |
| KungFuMaster | 22736 | 28999.80 | **42388.00** |
| MontezumaRevenge | **4367** | 154.00 | 0.00 |
| MsPacman | 15693 | 2570.20 | **18285.18** |
| NameThisGame | 4076 | **11686.50** | 8339.40 |
| Pong | 9.3 | **19.00** | 10.94 |
| PrivateEye | **69571** | 1704.40 | 100.00 |
| Qbert | 16455 | 18397.60 | **30618.50** |
| RoadRunner | 7845 | 54261.00 | **57212.00** |
| Robotank | 11.9 | **55.20** | 36.16 |
| Seaquest | **20182** | 19176.00 | 3922.80 |
| SpaceInvaders | 1652 | **12629.00** | 4289.40 |
| StarGunner | 10250 | **123853.00** | 20320.00 |
| Tennis | -8.9 | -2.20 | **0.00** |
| TimePilot | 5925 | 11190.50 | **24150.00** |
| Tutankham | 167.6 | 126.90 | **203.54** |
| Venture | 1188 | 45.00 | **1564.00** |
| VideoPinball | 17298 | **506817.20** | 139799.22 |
| WizardOfWor | 4757 | 14631.50 | **43436.00** |
| Zaxxon | 9173 | 19658.00 | **30818.00** |
| # Games >human | | 31 (67%) | **34 (74%)** |
| #Games >75% human | | 36 (78%) | **37 (80%)** |
| # Games Best | 7 (15%) | **21(46%)** | 18 (39%) |

Table 6: Same as Table 5 but with Rainbow using random no-op starts.

| GAME | Human | Rainbow | N-CPL |
|---|---|---|---|
| Alien | 6875 | **9491.70** | 6943.40 |
| Amidar | 1676 | **5131.20** | 2118.32 |
| Assault | 1496 | **14198.50** | 3079.92 |
| Asterix | 8503 | **428200.30** | 48226.00 |
| Asteroids | **13157** | 2712.80 | 9152.80 |
| Atlantis | 29028 | **826659.50** | 119636.00 |
| BankHeist | 734.4 | **1358.00** | 709.00 |
| BattleZone | 37800 | 62010.00 | **153880.00** |
| BeamRider | 5775 | **16850.20** | 3560.88 |
| Bowling | **154.8** | 30.00 | 103.24 |
| Boxing | 4.3 | **99.60** | 86.40 |
| Breakout | 31.8 | **417.50** | 302.04 |
| Centipede | 11963 | 8167.30 | **62654.16** |
| ChopperCommand | 9882 | **16654.00** | 3786.00 |
| CrazyClimber | 35411 | **168788.50** | 91912.00 |
| DemonAttack | 3401 | **111185.20** | 10876.00 |
| DoubleDunk | -15.5 | -0.30 | **23.96** |
| Enduro | 309.6 | **2125.90** | 220.28 |
| FishingDerby | 5.5 | **31.30** | -7.62 |
| Freeway | 29.6 | **34.00** | 28.96 |
| Frostbite | 4335 | **9590.50** | 6508.00 |
| Gopher | 2321 | **70354.60** | 12539.60 |
| Gravitar | **2672** | 1419.30 | 2284.00 |
| Hero | 25763 | **55887.40** | 36099.80 |
| IceHockey | 0.9 | 1.10 | **26.96** |
| Kangaroo | 3035 | **14637.50** | 9202.00 |
| Krull | 2395 | **8741.50** | 4422.88 |
| KungFuMaster | 22736 | **52181.00** | 42388.00 |
| MontezumaRevenge | **4367** | 384.00 | 0.00 |
| MsPacman | 15693 | 5380.40 | **18285.18** |
| NameThisGame | 4076 | **13136.00** | 8339.40 |
| Pong | 9.3 | **20.90** | 10.94 |
| PrivateEye | **69571** | 4234.00 | 100.00 |
| Qbert | 16455 | **33817.50** | 30618.50 |
| RoadRunner | 7845 | **62041.00** | 57212.00 |
| Robotank | 11.9 | **61.40** | 36.16 |
| Seaquest | **20182** | 15898.90 | 3922.80 |
| SpaceInvaders | 1652 | **18789.00** | 4289.40 |
| StarGunner | 10250 | **127029.00** | 20320.00 |
| Tennis | -8.9 | **0.00** | **0.00** |
| TimePilot | 5925 | 12926.00 | **24150.00** |
| Tutankham | 167.6 | **241.00** | 203.54 |
| Venture | 1188 | 5.50 | **1564.00** |
| VideoPinball | 17298 | **533936.50** | 139799.22 |
| WizardOfWor | 4757 | 17862.50 | **43436.00** |
| Zaxxon | 9173 | 22209.50 | **30818.00** |
| # Games >human | | **37 (80%)** | 34 (74%) |
| #Games >75% human | | **38 (83%)** | 37 (80%) |
| # Games Best | 6 (13%) | **31(67%)** | 10(22%) |

Table 7: Mean with standard deviations of 20 RTDP vs Random episodes for the SMRF test. Bold domains are classified as having SMRF.

| GAME | RTDP | Random |
|---|---|---|
| Alien | 379.50 ± 180.1 | 135.00 ± 39.4 |
| Amidar | 30.90 ± 18.0 | 8.70 ± 15.8 |
| Assault | 139.65 ± 38.3 | 49.35 ± 27.6 |
| Asterix | 625.00 ± 91.5 | 155.00 ± 70.5 |
| Asteroids | 397.50 ± 137.6 | 74.00 ± 42.0 |
| Atlantis | 1350.00 ± 915.7 | 635.00 ± 979.9 |
| BankHeist | 27.50 ± 20.7 | 3.50 ± 4.8 |
| BattleZone | 3350.00 ± 2725.3 | 450.00 ± 669.0 |
| BeamRider | 121.00 ± 30.7 | 50.60 ± 46.7 |
| Berzerk | 324.00 ± 109.4 | 102.50 ± 68.0 |
| **Bowling** | 0.00 ± 0.0 | 0.00 ± 0.0 |
| Boxing | 12.90 ± 5.8 | −2.00 ± 4.1 |
| Breakout | 3.10 ± 0.8 | 1.00 ± 0.7 |
| Centipede | 2189.75 ± 226.5 | 548.85 ± 565.0 |
| ChopperCommand | 790.00 ± 260.6 | 205.00 ± 120.3 |
| CrazyClimber | 715.00 ± 127.6 | 395.00 ± 124.4 |
| DemonAttack | 68.50 ± 9.6 | 29.50 ± 12.4 |
| DoubleDunk | −0.20 ± 1.4 | −1.90 ± 1.3 |
| **Enduro** | 0.15 ± 0.5 | 0.05 ± 0.2 |
| FishingDerby | −1.70 ± 3.1 | −9.70 ± 3.0 |
| **Freeway** | 0.00 ± 0.0 | 0.00 ± 0.0 |
| Frostbite | 122.00 ± 21.1 | 23.50 ± 16.5 |
| **Gopher** | 0.00 ± 0.0 | 0.00 ± 0.0 |
| Gravitar | 177.50 ± 195.2 | 0.00 ± 0.0 |
| Hero | 2206.50 ± 886.9 | 36.25 ± 41.4 |
| IceHockey | 0.80 ± 0.8 | −0.40 ± 0.8 |
| **Jamesbond** | 15.00 ± 22.9 | 5.00 ± 15.0 |
| Kangaroo | 100.00 ± 100.0 | 10.00 ± 43.6 |
| Krull | 104.50 ± 38.4 | 19.50 ± 18.8 |
| KungFuMaster | 130.00 ± 145.3 | 5.00 ± 21.8 |
| **MontezumaRevenge** | 0.00 ± 0.0 | 0.00 ± 0.0 |
| MsPacman | 396.50 ± 193.9 | 213.00 ± 163.7 |
| NameThisGame | 138.00 ± 41.1 | 44.50 ± 35.3 |
| Phoenix | 346.00 ± 93.0 | 110.00 ± 66.8 |
| **Pitfall** | 0.00 ± 0.0 | −1.10 ± 4.8 |
| Pong | −0.65 ± 1.4 | −3.30 ± 1.0 |
| **PrivateEye** | 0.00 ± 0.0 | 4.75 ± 21.9 |
| Qbert | 606.25 ± 50.5 | 77.50 ± 125.5 |
| Riverraid | 862.50 ± 210.2 | 321.50 ± 70.0 |
| **RoadRunner** | 15.00 ± 65.4 | 5.00 ± 21.8 |
| Robotank | 1.05 ± 0.7 | 0.30 ± 0.5 |
| Seaquest | 73.00 ± 26.3 | 14.00 ± 15.6 |
| **Skiing** | −1248.00 ± 0.0 | −1248.00 ± 0.0 |
| **Solaris** | 0.00 ± 0.0 | 0.00 ± 0.0 |
| SpaceInvaders | 130.75 ± 29.5 | 41.00 ± 25.6 |
| StarGunner | 370.00 ± 95.4 | 60.00 ± 106.8 |
| Tennis | 0.10 ± 0.7 | −1.90 ± 0.9 |
| TimePilot | 370.00 ± 134.5 | 105.00 ± 66.9 |
| Tutankham | 17.85 ± 11.8 | 1.95 ± 4.8 |
| UpNDown | 1247.50 ± 465.4 | 272.00 ± 285.5 |
| **Venture** | 0.00 ± 0.0 | 0.00 ± 0.0 |
| VideoPinball | 5026.10 ± 1909.3 | 521.10 ± 365.4 |
| WizardOfWor | 240.00 ± 156.2 | 30.00 ± 55.7 |
| YarsRevenge | 5527.45 ± 2895.4 | 274.10 ± 315.1 |
| **Zaxxon** | 20.00 ± 60.0 | 0.00 ± 0.0 |

## Additional Implementation Details

The code along with the experiment result files are available at https://github.com/stefanotoole/N-CPL.

Table 8: N-CPL hyperparameters.

| Lookahed Parameters | |
|---|---|
| Sim. Interactions Budget | 100 |
| Lookahead Horizon | 100 |
| **Value and Policy Network Parameters** | |
| Batch size | 128 |
| Learning Rate | $2.50 \times 10^{-4}$ |
| Epochs | 8 |
| Loss Function for Policy | Categorical crossentropy |
| Loss Function for Value Function | Huber |
| Discount factor used in TD Learning | 0.99 |
| Time steps between target network updates (for value network) | 10,000 |
| Interval size of learning schedule | $1 \times 10^6$ sim. interactions |

Table 8 shows the different hyperparameters of N-CPL along with the selected values used for the experiments. Due to computational restraints we could not tune the hyperparameters of N-CPL. For the lookahead parameters the simulator budget per time step was selected to be 100 to match the budget used by Junyent et al. [1]. The lookahead horizon is the maximum search depth allowed, that is the maximum number of actions allowed from the root node of the lookahead. For the value and policy network parameters both the batch size and number of epochs used affects the training time. We selected both the batch size and number of epochs such that the time spent training the networks using a single vCPU is around the same time as the planning steps of N-CPL. We used the same learning rate, loss function (for the value function) and discount factor for the TD learning as used by DQN [3]. Following Junyent et al.[5] we use a crossentropy loss function for the policy network. The interval size of the learning schedule dictates the size of the data set used to update the networks and how often to reject or accept parameter updates. The interval size of the learning schedule is illustrated in Figure 2 of the main paper to be of a size of N episodes. Instead of setting the interval size to be a set number of episodes we set it as $1 \times 10^6$ simulator interactions.