# OpenReview forum: "Width-based Lookaheads with Learnt Base Policies and Heuristics Over the Atari-2600 Benchmark"
_NeurIPS.cc/2021/Conference — NeurIPS 2021 Spotlight_

### Official Review · Reviewer_Ae7H · 2021-06-25

**Rating:** 6
**Confidence:** 4

**Summary:**

The paper builds upon existing width-based methods for solving Atari games environment, in particular Rollout-IW, and enhances them with Critical Path Value Functions to achieve better results.

**Limitations And Societal Impact:**

Yes

**Main Review:**

Novelty:
Although the novelty based rollouts and the critical paths are not novel themselves, the application of critical paths to enhance width-based search as presented in the paper is novel.

Clarity:
The paper is written in a somewhat confusing manner. My main issue is that the RIW+CPV (algorithm constituting the main contribution of the paper) is presented in a very informal way, thus it is hard for the reader to comprehend how exactly does it function. Also, some sentences are structured in a very confusing way and some decisions are left unexplained.

Scholarship:
The literature review is sufficient.
It should be noted, however, that you omitted the work of Dittadi et al. from section 4.2, where you discuss the novelty features. Their representation of the features was shown to perform better than B-PROST, and you mention them in the introduction, so it is unclear why this work uses B-PROST.

Reproducibility:
I was not able to launch the code by following the instructions in the readme file. However, this might be due to my system’s limitations.

Notes:
Line 34: “... we consider planners which use novelty measures directly over the values of screen pixels” - no justification is given for this decision.

Line 61: “... we consider [...] deterministic action transitions” - Atari screen setting is not deterministic as far as I’m aware (as opposed to Atari RAM setting)

Line 65: It doesn’t make sense to me to draw from T if you assume deterministic action transitions.

Line 66: Why introduce \gamma just to say that it's 1?

Def. 1 (lines 75-77): What do you mean by “nodes/states”? Does it mean that there are 2 definitions here, one for lookahead tree with states and the other one for lookahead tree with nodes? Or do you declare that here states and nodes are two different names for the same type of objects? What is the connection between these and the MDP described above?

Lines 78-79: In section 2.1 the rewards are defined on pairs (a, s), where s is a state but here you mention “rewards for each node”, please clarify.

Line 90: Again, it doesn’t make sense to me to draw from T if you assume deterministic action transitions

Lines 155-156: This sentence confused me. Do RIW(1) and RIW_{D} refer to the same algorithm? If so, why give it a different name?

Lines 190-201: The description of the algorithm is very informal. I would suggest formalizing it, or at least attach a graphics to help the reader understand the process

In the result tables - it's not clear whether the number in row $i$ column $j$ represents how many times $i$ beats $j$ or $j$ beats $i$.

Lines 359-360: Is the taxonomy that you refer to just the binary quality of a game being SMRF or not?

Summary:
The main idea of the paper looks promising, but sadly the clarity of this paper is subpar.


**Time Spent Reviewing:**

1

---

> ### Author Response · Authors · 2021-08-09
> **Thank you for your review.**
>
> Thank you for your review and useful critiques of our work. We are glad that you appreciate the ideas presented despite your concerns around clarity. We are absolutely sure that through incorporating the suggestions from each of these reviews we can address your clarity concerns. In particular, your suggestion for improving the algorithm description, Rev. 1’s suggestion around describing the learnt value function and Rev. 2’s recommendation around the phrasing of Section 4.
>
> For your comments about omitting a description of Dittadi’s paper and using BPROST features instead. Our work does not use the BPROST feature set - please see the last paragraph of 4.2.  We do agree it would be nice to include a discussion about Dittadi's representation of features and its complexity compared with our method of using the gray scale pixel values directly. Also please note that Dittadi's representation of features reported still used the BPROST feature set in order to collect training data. We will add a short mention of their method in Section 4.2 for the Camera-Ready version.
>
> In regards to the reproducibility, we did test running the code on a fresh VM install following the readme instructions before submitting. We will also be making the repo public on github, so if anyone has any issues running the experiments they can post an issue and we can help them fix it. We should also note that we only tested the code using python 3.6.9 (as mentioned in the readme) and on a Ubuntu 18.04 LTS system (we will add this note to the repo). If you wish to let us know the error you were receiving we are happy to try and help you fix the issue through the Author Discussion period.
>
> Thanks for pointing out the need for justification on line 34. The justification for using a novelty measure directly over the pixels is that it is simpler, that is, it does not require feature engineering which closes the gap of comparison with other learning methods that also rely only on screen features. We will change this sentence to "In this paper we consider planners with a simplified novelty measure that does not require extensive feature engineering or the internal state of the simulator but is instead defined directly over the values of screen pixels.".
>
> In line with the previous work [1, 2, 3],  we use the internal game state for the state-action transitions such that they are deterministic. However for our feature set and for learning we only use the observable screen pixel values, we will add clarifying statements about this into Sections 2.1 and 4.
>
> Problems that have a deterministic transition function are still MDPs, they are just a special case. Including T and gamma allow us to specify that we are dealing with a special case of MDPs.
>
> You are correct the “states/nodes” statement is confusing. We will fix this to just be nodes, defining a node to be a state-action path starting at the initial state of the lookahead. Line 78 will also be changed to “Through backing up the rewards for each node in the lookahead an expected..” and we will change nodes to be ‘n’ in the notation to distinguish between nodes and states. Finally where any of the definitions require the state, $s$ a node, $n$ refers to, that is the last state in the state-action path the node represents, we will introduce and refer to $s$ as $n^s$. For example Definition 2 will change to:
>
> Given a lookahead L= ($N$, $C$, $n_r$) (Definition 1) the action to execute $a$ is selected at the root node $n_r$ by $argmax_{a \in A(n_r^s)}\\{ Q(n_r, a)\\}$, where $Q(n_r,a) = R(a, n_r^s) + \sum_{n \in C(n_r, a)} T(n_r^s, a, n^s) V(n)$, where $V(n) = V^T(n^s)$, with $V^T(n^s)$ being a termination cost, when $C(n) = \emptyset$, otherwise $V(n) = max_{a \in A(n^s)} \\{R(a, n^s) + \sum_{n' \in C(n, a)} T(n^s, a, n'^s)V(n')\\}$.
>
> Yes, RIW(1) is the same as RIW$_D$, the different name is so that it is clear that the only difference between RIW$_D$ (RIW (1)) and RIW$_C$ is their novelty definition. We will rephrase that sentence to “In what follows, we refer to Bandres et al.’s original configuration of RIW(1) as RIW$_D$ and refer to RIW(1) where one replaces the "Depth" novelty definition with the "Classic" one as RIW$_C$.”.
>
> Yes, the taxonomy is for games having larger vs smaller branching factors and being SMRFs or not.
>
> We’d like to make you aware, for future reviews, that in your time spent response the identity of the person you collaborated with for the review has been revealed.
>
> [1] Junyent, Miquel, Vicenç Gómez, and Anders Jonsson. "Hierarchical Width-Based Planning and Learning." Proceedings of the International Conference on Automated Planning and Scheduling. Vol. 31. 2021.
>
> [2] Junyent, Miquel, Anders Jonsson, and Vicenç Gómez. "Deep policies for width-based planning in pixel domains." Proceedings of the International Conference on Automated Planning and Scheduling. Vol. 29. 2019.
>
> [3] Bandres, Wilmer, Blai Bonet, and Hector Geffner. "Planning with pixels in (almost) real time." Proceedings of the AAAI Conference on Artificial Intelligence. Vol. 32. No. 1. 2018.

---

> > ### Comment · Reviewer_Ae7H · 2021-08-09
> > **Update**
> >
> > Given the detailed response by the authors, I am convinced they will address the main issues with clarity in the final version, and so will upgrade my score.

---

### Official Review · Reviewer_RtGM · 2021-07-16

**Rating:** 8
**Confidence:** 4

**Summary:**

  The paper proposes a new state of the art in planning-based approach
  to Atari game.  The paper carefully analyses existing work, including
  $pi$-IW+ which uses both value and policy functions, and fixes several
  incorrect decisions that qualitatively does not make sense, such as
  mixing up the estimate for the base and the critical path policy.

  Perhaps what I would ask the authors is the way the paper is written:
  The current text is not written in the style of "They do X which has
  this issue; therefore we do Y" --- issue-focused description, but in
  the style of "We do Y; X does this; X is problematic".  For example,
  Sec 4.2 could be titled as "Addressing the complex novelty definition
  in RIW", or "Simplifying the novelty; depth-based novelty is not
  necessary".

  The algorithm also has a mechanism inspired by AlphaZero, which
  resembles active-learning (section 4.5), which compares the old and
  the updated parameters.  Following up work should more carefully
  evaluate this mechanism as an active-learning scheme.

  Its second contribution is the proposal of a standard evaluation
  scheme, which is now an adequate proposal due to the recent
  improvement in the field.

  The proposed algorithm is carefully evaluated empirically,
  characterizing the sparseness and the branching factor.

  This is a solid paper that I recommend for acceptance.


**Limitations And Societal Impact:**

The paper deals is categorized as basic research rather than applied research. It is not likely to cause large societal issue.
The evaluation is also performed with much less compute than what is typically used in RL papers, and is likely to have less environmental impact.

**Main Review:**

  Throughout the paper, "base policy" could be replaced with "uniform
  random policy" to make the meaning immediately obvious.  In the
  current text, readers should trace the meaning back to line 120-121,
  and it may still be a bit ambiguous.

  l.134, and are hard to distinguish, consider alternative punctuations.
  Well, shouldnt you just use V if and V are same?


  There are quite a lot of variations of the algorithms.  I suggest
  adding a feature comparison table between them.

  The table row / column ordering in Table 1,2 could be reconsidered to
  emphasize your proposed approach --- RIWD+CPB at the top.

  There are inconsistencies in the references (conference names, etc.)

  Regarding the full vs minimal action sets, [1] [2] are worth cited.



  [1] Jinnai, Yuu, and Alex Fukunaga. "Learning to prune dominated
  action sequences in online black-box planning." Proceedings of the
  AAAI Conference on Artificial Intelligence. Vol. 31. No. 1. 2017.

  [2] Mark J. Nelson (2021). "Estimates for the branching factors of
  Atari games." In Proceedings of the IEEE Conference on Games.


**Time Spent Reviewing:**

2.5

---

> ### Author Response · Authors · 2021-08-09
> **Thank you for your comments.**
>
> We greatly appreciate your effort invested into the review and the helpful suggestions you provided.
>
> We think your suggestion for the change in style for Section 4 is excellent, especially the example titles you gave for the subsections. We will definitely change the text and subheadings to match this style.
>
> In regards to the base policy, it can be any policy, just the original RIW(1) defines it as a uniform random policy, for example, RIW+CPV defines the base policy through a NN. We will amend the text to try to make this clearer.
>
> Agree that line 134 could be confusing, we will fix it.
>
> We’ll reorganise the table as suggested to have the following order: RIW_C+CPV, RIW_D+CPV, RIW_C, RIW_D, \pi-IW(1), \pi-IW(1)+, \pi-HIW(n,1).
>
> We’ll fix any inconsistencies in the references.
>
> Thanks for suggesting the two citations for minimal vs full action sets, we will add them in, also pointing out that as was shown in [2] the true average branching factor is actually much smaller than the minimal action set and learning the minimal action set as in [1] could help avoid “wasted” calls to the simulator.

---

> > ### Comment · Reviewer_RtGM · 2021-08-10
> > **One additional question regarding the benchmark setting**
> >
> > Hi, thanks for the answers.
> >
> > I would like to ask one more question regarding the benchmark setting: In line 296, you mentioned that the simulator call budget is not increased within an episode (multiple time steps, each with multiple rollouts) if the node already exists in the tree.
> > What happens between episodes, i.e., after a reset (e.g., the agent dies, in-game clock runs out, or game is solved) ?
> > If the same state is visited by the same sequence of actions after the episode, is it counted or not counted?
> >
> > And I am sorry, my comments about line 134 are corrupted when I exported from emacs org mode to a text file. It is about $\bar{V}$ and $\tilde{V}$, which were not easy to distinguish visually.

---

> > > ### Author Response · Authors · 2021-08-10
> > > **Lookahead transitions are only cached within an episode**
> > >
> > > Between episodes the lookahead is cleared and reset. If a transition is visited by the same state-action path as visited in a previous episode but has not been visited in the current episode a simulator call is required and counted. To make this clearer we will change line 296 to:
> > >
> > > "That is, when the search revisits a transition between two nodes of the lookahead within the same episode, the simulator does not need to be recalled and hence does not affect the simulator budget."
> > >
> > > No worries about your line 134 comment we could decipher what you meant.

---

> > > > ### Comment · Reviewer_RtGM · 2021-08-14
> > > > **Error in the Detailed number**
> > > >
> > > > I found two error: In line 286, you wrote "3000 simulator call with frameskip of 5", however, it should be 30000.
> > > > Also, in line 283, 2x10^7 is not the same as Junyent et al; it should be noted that the value was changed.
> > > >
> > > > Lipovetzky et al , 2015 says "Our experimental setup follows theirs except that a maximum budget of 150, 000 simulated frames is applied to IW(1), 2BFS, and UCT" in page 3 (which is ambiguous) and "a lookahead budget of 150,000 simulated frames" in page 5. This means 30,000 nodes=actions in each lookahead=planning, i.e., 30000 nodes between each acting.
> > > >
> > > > It may also be worth noting that Bellmere et al used 133000 frames between each action in BrFS.
> > > >
> > > > Junyent et al ICAPS2019 says "Performance is measured after 40M generated nodes, i.e., interactions with the simulator (excluding skipped frames)" with frameskip of 15. This amounts to 600M frames, or 6x10^8 frames.
> > > >
> > > > It is Junyent et al ICAPS2021 that says " the same setup as in Junyent, Jonsson, and Gómez (2019), but half the budget of simulator interactions" , that is, 20M interactions, or 300M frames.
> > > >
> > > > Also, in the appendix, Table 3 says 20x10^7, but it should be actually 2x10^7 = 20 x 10^6 = 20M.
> > > > Please recheck the numbers for DQN / rainbow.
> > > >
> > > > I started to feel the word "simulator call" is very ambiguous, and since the consistent, reproducible evaluation setting is a contribution in this paper, I urge the authors to review and clarify the terminology. Even if the frames are skipped, we still need to send messages to the simulator to skip the frame while pressing the same button --- That is also a "call". What I believe is unambiguous is "number of frames", including all skipped frames.

---

> > > > > ### Author Response · Authors · 2021-08-16
> > > > > **Clarification of typos and response regarding the simulator calls terminology**
> > > > >
> > > > > Thanks for pointing out those details and continuing to help us improve the paper.
> > > > >
> > > > > You are correct that both the 3,000 in line 286 and $10^7$s in Table 3 of the appendices are typos. In line 19 of the appendices, it should also be $10^6$ not $10^7$.
> > > > >
> > > > > Yes, you are also correct about the budgets used in Junyent et al. ICAPS 2019 and 2021 [2,3], but please note that in lines 282-283 we state that we use Junyent et al. ICAPS 2021’s [3] evaluation settings not the settings from Junyent et al. ICAPS 2019 [2].
> > > > >
> > > > > The frameskip is a setting of the environment, not the agent. As we state in each of the titles of the DQN and Rainbow comparison tables (in the appendices), the experimental settings are not the same as RIW+CPV, with one reason being the different frame skip used. The number of frames used including the skipped frames can also be ambiguous in a number of different ways:
> > > > > 1. The DQN paper [1] reports the frames excluding frame skips, please see the Methods appendix under the Training details “We trained for a total of 50 million frames (that is, around 38 days of game experience in total)”. Atari runs at 60 fps, so converting the frames to the equivalent game experience time we get 50,000,000/(60 fps * 60 seconds per minute * 60 minutes per hour * 24 hours per day) ~= 9.65 days, given that they used a frameskip of 4 we can see they did not include the skipped frames in the 50 million frames budget as 38/9.65 ~= 4.
> > > > > 2. The number of frames (including the skipped frames) is not equivalent to the number of frames seen by the agent. The agent does not get to observe the skipped frames. When stacking the 4 most recent frames for the NN input (in DQN [1], Junyent et al. 19 and 21 [2, 3], and our paper) these are the 4 most recent frames observed by the agent, that is they do not include the skipped frames.
> > > > > 3. Given the same budget of frames (including the skipped frames), learning methods with smaller frame skips will have more training data.
> > > > > 4. Our paper is focused on lookahead methods and the number of simulator calls equals the number of nodes generated by the lookaheads. While using the number of frames including the skipped frames does not indicate how many nodes are considered by the lookaheads.
> > > > >
> > > > > Hence, we avoid using the frames terminology and instead use simulator calls. We also explicitly state whenever comparing to other algorithms that the frame skip used by the environment changes the experimental setting (lines 271-273, Appendix Table 3,4,5). In line 67, we define the simulator of the environment. We do agree with your point that the simulator call terminology may be ambiguous so we will change it to the number of environment interactions as used by [2, 3].
> > > > >
> > > > > [1] Mnih, Volodymyr, et al. "Human-level control through deep reinforcement learning." nature 518.7540 (2015): 529-533.
> > > > >
> > > > > [2] Junyent, Miquel, Anders Jonsson, and Vicenç Gómez. "Deep policies for width-based planning in pixel domains." Proceedings of the International Conference on Automated Planning and Scheduling. Vol. 29. 2019.
> > > > >
> > > > > [3] Junyent, Miquel, Vicenç Gómez, and Anders Jonsson. "Hierarchical Width-Based Planning and Learning." Proceedings of the International Conference on Automated Planning and Scheduling. Vol. 31. 2021.

---

> > > > > > ### Comment · Reviewer_RtGM · 2021-08-27
> > > > > > **Thanks**
> > > > > >
> > > > > > Thanks for the clarifications. I also feel "environment interactions" is better.
> > > > > >
> > > > > > Re: frame, it is still a matter of debate, though... For the real applications, what matter is the real frame including the skipped frames. It is obvious if the "simulator" is the real environment. If, instead of Atari, the simulator is one for a large industrial / chemical plant, implemented with an expensive-to-compute and precise physics CAD/CFD model, then every frame probably counts. The fact that we skip frames is our arbitrary decision to keep up with the runtime, and I don't think we should claim it as if skipping frames is an advantage. In other words, we don't "do something with fewer data" just by skipping frames.

---

> > > > > > > ### Author Response · Authors · 2021-08-30
> > > > > > > **Thanks for the discussion points**
> > > > > > >
> > > > > > > Thanks again for the great discussion points, they are spot on.
> > > > > > >
> > > > > > > Let us clarify that we were not trying to claim that skipping frames is an advantage, we were rather just pointing out the ambiguities and implications of using the number of frames including the skipped ones. We think the least ambiguous way of describing the experimental setting is with the full context of the environment interactions + the frameskip used. We also agree 100% with the implications you point out about not including the skipped frames. In fact, it is also possible to consider other applications where it could be faster to compute frames that are not sent to the agent as they won't require rendering, and it would be enough to just update the internal state of the simulation, or perhaps even increase the time step of the simulation to skip the computation of those states altogether.

---

### Official Review · Reviewer_XEoW · 2021-07-16

**Rating:** 7
**Confidence:** 4

**Summary:**

This paper introduces a novel width-based algorithm, RIW+CPV, for playing Atari 2600 games. RIW+CPV is based on RIW, an algorithm that performs a depth-first search while pruning nodes according to IW's novelty metric and it guides the search with a base policy (later works showed how to learn such a base policy). In addition to the base policy, RIW+CPV learns a value function that is used to evaluate the leaf nodes of the lookahead tree RIW expands before deciding on the action that will be applied in the environment. Empirical results show that RIW+CPV achieves higher scores on a larger number of games than other IW-based methods for Atari 2600 games.

**Limitations And Societal Impact:**

Yes, the authors have addressed the limitations of their study and proposed algorithm, such as the lack of ablation studies.

**Main Review:**

The paper lists the following contributions (see last paragraph of Section 1):

(1) analysis of previous IW algorithms for Atari games;
(2) new IW-based algorithm
(3) learning schedule
(4) identifying characteristics of Atari 2600 games that influence the performance of IW-based planners

Contributions (2) and (3) are in fact a single contribution. The new search algorithm uses a learning schedule for updating the weights of the neural models responsible for encoding the policy and value functions. In my opinion (2)+(3) is the main contribution of the work. The new algorithm, RIW+CPV, learns a value function that is used to evaluate the leaf nodes of the lookahead tree the algorithm expands before deciding on the next action the agent will take.

The original RIW algorithm used V-values of zero for all leaf nodes (i.e., RIW assumes that the agent won't receive any reward from a leaf state onwards). This is a reasonable assumption to make as long as the lookahead tree is able to "look far enough" into the future. The problem is that in (almost) real-time search one is unable to look far enough into the future. The present work replaced the V-value of zero by a value function that estimates the expected reward value from the leaf node.

The CPV in RIW+CPV stands for Critical Path Value Function. The critical path is formed by the set of actions the algorithm chooses after completing its search in each "decision point." The paper explains that the value function is learned for the critical path and not for the base policy that is used to bias the lookahead tree. I believe there is a better way of describing the value function RIW+CPV learns. The value function returns a value that estimates the expected reward if a given policy is used from a given state onwards. The paper correctly explains that the base policy isn't the value function's policy. The lookahead search can be seen as a policy improvement operator, similar to how PUCT is to AlphaZero. The value function RIW+CPV is in fact for the policy induced by this  operator. I would define this policy (perhaps \pi_{IW}?) and explain that the value function learned is for \pi_{IW}.

RIW+CPV is indeed stronger than previous IW methods as shown in the experiments. However, I missed in the experiments a baseline that doesn't use novelty. RIW+CPV is similar to AlphaZero in the sense that it employs both a policy and value function to guide its search. One then wonders how much of the results are explained by IW's clever pruning or by the learned policy and value functions. It would be nice if the authors could include results of a version of RIW+CPV in which the IW part is turned off.

Another suggestion is to use the learned value function to refine the novelty function. The paper considers two novelty definitions: classic and depth-based. However, it misses a definition that uses ideas from the classical planning literature where a heuristic function is combined with novelty. For example, [1] uses novelty to break ties of nodes with the same h-value. The novelty values in the context of Atari 2600 games could be refined with the V-values by considering a node n novel according to feature f even if another node n' had already satisfied the novelty condition with respect to f as long as V(n) is different than V(n').

Still under the umbrella of the contribution (2)+(3), I enjoyed the fact that the novelty is computed in terms pixels in this work. It is a pity that the paper doesn't present ablation studies to help us understand whether to good performance comes from the novelty representation or from the learned value function.

The contribution (1) is fair. I enjoyed reading the related work section of the paper. I was already somewhat familiar with the literature though. I wonder if someone new to the field will also appreciate the analysis of previous algorithms. The analysis would be better if it had explained how the hierarchical \pi-HIW(n, 1) relates to the current work. For example, can one apply the ideas introduced in this work to the hierarchical algorithm?

I was somewhat confused with the experiments of contribution (4). This is mainly because the tables of results do not report a normalized average win rate, but only the total number of wins. It is very hard to compare across tables without a normalized value.

OTHER SUGGESTIONS

In the conclusion it reads "We hope that by having addressed some of the discrepancies in the experimental settings of previous width-based algorithms, such as the size of the action set and the planning budget used, [...]" I disagree that this paper addresses the discrepancies. This paper acknowledges the discrepancies and chooses one of the options that had been used in the past. The paper doesn't explain why the chosen option is superior and should be the standard option for future works.

"without loss of generality, we consider only discrete action sets and deterministic action transitions." I disagree that this is without loss of generality. It isn't clear how RIW works with continuous action and state spaces and how it deals with stochastic transitions.

Reference:

[1] Nir Lipovetzky and Hector Geffner. Best-first width search: Exploration and exploitation in classical planning. In AAAI Conference on Artificial Intelligence, pages 3590–3596, 2017.

**Time Spent Reviewing:**

6

---

> ### Author Response · Authors · 2021-08-09
> **Thank you for your review. We provide additional results for a version of RIW+CPV with novelty turned off and address your other comments.**
>
> Thank you for the time you invested into your thorough review of our paper.
>
> You provide an excellent summary of the issue with using a V-value of zero for leaf nodes within Rollout-IW’s lookahead. We would like to note that zero V-values on leaf nodes is problematic as well for shortest path problems. As we mention in Section 4.3 using cost-go-approximations at leaf nodes has been shown to significantly improve the performance of width-based lookaheads over shortest path problems.
>
> We find your suggestion of how to describe the Value function learned by RIW+CPV to be very valuable. We will ensure to incorporate your suggestion into the Camera-Ready version.
>
> In terms of including a version of RIW+CPV with novelty turned off, we agree it would be helpful to include this into the result tables. Due to computational constraints we can not provide a full ablation study of every design decision within RIW+CPV. However, we agree that by adding RIW+CPV with novelty turned off we would have a complete study on the most important mechanisms of RIW+CPV. That is, learning verse not learning (RIW vs RIW+CPV), novelty definition (RIW$_D$+CPV vs RIW$_C$+CPV) and pruning for novelty verse not as you describe.  We have started to run RIW+CPV in that configuration. At the time of writing this we can only report results for ⅕ of trials. We will have the results for all the trials included in the Camera-Ready version. The updated preliminary results for the total number of times each algorithm is the best algorithm in the pairwise comparisons for the overall benchmark set are (the version with novelty turned off we have as R+CPV):
>
> Full benchmark set:
>
> | Algorithm            | Total Wins  |
> |----------------------|-------------|
> | RIW$_C$+CPV            | 258 (69.5%) |
> | RIW$_D$+CPV            | 250 (67.4%) |
> | R+CPV                | 215 (58.0%) |
> | RIW$_C$                | 95  (25.6%) |
> | RIW$_D$                | 104 (28.0%) |
> | $\pi$-IW(1)            | 184 (49.6%) |
> | $\pi$-IW(1)+           | 164 (44.2%) |
> | $\pi$-HIW(n,1)         | 199 (53.6%) |
>
> We can see that the method with novelty turned off (R+CPV) is still outperformed by RIW$_C$+CPV and RIW$_D$+CPV, however it does outperform all other methods.
> We would also like to point out that we do provide a comparison of our results against model-free methods that do not utilise novelty in the Appendices.
>
> Thanks for your suggestion about refining the novelty definition using the learnt value function this is something that we will definitely include in future work.
>
> In response to your comment about including an ablation study to see if the good performance comes from the novelty feature representation or the learning of CPV, the results for both RIW_D and RIW_C do use the same feature representations for their novelty definitions as the learning methods RIW$_D$+CPV and RIW$_C$+CPV, so we can at least see the benefit CPV provides. Additionally with the R+CPV version we mention above, and which will be included in the final version of the paper, we can see the benefit of using novelty as well. As mentioned in lines 260-270 it would be unfair for us to compare our results of RIW$_D$ and RIW$_C$ directly with the original results on Atari for RIW presented by Bandres et al. [1] as we evaluate on Atari using the minimal action set of each game, as done by the $\pi$-IW variants.
>
> Thank you for pointing out that it was hard to compare the results across the different tables without a normalised win-rate value. For the Camera-Ready version we will include the percentage win rate along with the absolute win rate. For example for the overall win rates of each algorithm:
>
> Full benchmark set:
>
> | Algorithm    | Total Wins  |
> |--------------|-------------|
> | RIW$_C$+CPV    | 225 (70.8%) |
> | RIW$_D$+CPV    | 220 (69.2%) |
> | RIW$_C$        | 84  (26.4%) |
> | RIW$_D$       | 95  (29.9%) |
> | $\pi$-IW(1)    | 160 (50.3%) |
> | $\pi$-IW(1)+   | 145 (45.6%) |
> | $\pi$-HIW(n,1) | 174 (54.7%) |
>
>
> Branching Factor >=10:
>
> | Algorithm    | Total Wins  |
> |--------------|-------------|
> | RIW$_C$+CPV    | 146 (73.7%) |
> | RIW$_D$+CPV    | 149 (75.3%) |
> | RIW$_C$        | 63  (31.8%) |
> | RIW$_D$        | 67  (33.8%) |
> | $\pi$-IW(1)    | 84  (42.4%) |
> | $\pi$-IW(1)+   | 70  (35.4%) |
> | $\pi$-HIW(n,1) | 104 (52.5%) |
>
>
> SMRF games:
>
> | Algorithm    | Total Wins |
> |--------------|------------|
> | RIW$_C$+CPV    | 50 (69.4%) |
> | RIW$_D$+CPV    | 44 (61.1%) |
> | RIW$_C$        | 22 (30.5%) |
> | RIW$_D$        | 21 (29.2%) |
> | $\pi$-IW(1)    | 25 (34.7%) |
> | $\pi$-IW(1)+   | 37 (51.4%) |
> | $\pi$-HIW(n,1) | 43 (59.7%) |
>
> We agree with your statement regarding the wording used in the conclusion. As we suggest in the discussion section, future work should look to set out an evaluation protocol for benchmarking planning algorithms on the Atari-2600 games. We found that the previous works would ignore these discrepancies all together so we will change the wording from “addressing” the discrepancies to "We hope that by having identified..".
>
> We will also amend the text to remove the “without loss of generality”, you are correct that it is unclear how RIW would work with stochastic transitions, and continuous action and state spaces.
>
> [1] Bandres, Wilmer, Blai Bonet, and Hector Geffner. "Planning with pixels in (almost) real time." Proceedings of the AAAI Conference on Artificial Intelligence. Vol. 32. No. 1. 2018.

---

> > ### Comment · Reviewer_XEoW · 2021-08-26
> > **thanks for the detailed response**
> >
> > Thank you for your detailed response and for the additional results. Please include them in the final version of the paper.
> >
> > Also, it is so much easier to read the tables with the added win rate column. Thank you!

---

### Author Response · Authors · 2021-08-09
**General response to all reviewers**

We thank all the reviewers for their positive feedback about our work and identifying the impacts of our contributions and results presented in the paper. We would also like to thank the reviewers for providing suggestions about how we can improve the clarity and presentation of our work. We believe all the suggestions are straightforward minor presentation changes and we will ensure to thoroughly address them for the camera ready version. We provide direct responses to each reviewer to address any questions or misunderstandings, along with exactly how we intend to address any of their concerns.

---

### Decision · Program_Chairs · 2021-09-27

**Decision:**

Accept (Spotlight)

**Comment:**

The reviewers have written careful reviews and are generally in agreement that the paper is making a nice contribution. The authors are strongly encouraged to revise the final paper in accordance with the substantial feedback from the reviewers.